# A computationally informed comparison between the strategies of rodents and humans in visual object recognition

**Anna Elisabeth Schnell[1]\*, Maarten Leemans[1], Kasper Vinken[2], Hans Op de Beeck[1]**

[1]Department of Brain and Cognition & Leuven Brain Institute, Leuven, Belgium;
[2]Department of Neurobiology, Harvard Medical School, Boston, United States

**Abstract** Many species are able to recognize objects, but it has been proven difficult to pinpoint and compare how different species solve this task. Recent research suggested to combine computational and animal modelling in order to obtain a more systematic understanding of task complexity and compare strategies between species. In this study, we created a large multidimensional stimulus set and designed a visual discrimination task partially based upon modelling with a convolutional deep neural network (CNN). Experiments included rats (N = 11; 1115 daily sessions in total for all rats together) and humans (N = 45). Each species was able to master the task and generalize to a variety of new images. Nevertheless, rats and humans showed very little convergence in terms of which object pairs were associated with high and low performance, suggesting the use of different strategies. There was an interaction between species and whether stimulus pairs favoured early or late processing in a CNN. A direct comparison with CNN representations and visual feature analyses revealed that rat performance was best captured by late convolutional layers and partially by visual features such as brightness and pixel-level similarity, while human performance related more to the higher-up fully connected layers. These findings highlight the additional value of using a computational approach for the design of object recognition tasks. Overall, this computationally informed investigation of object recognition behaviour reveals a strong discrepancy in strategies between rodent and human vision.

\*For correspondence:
annaelisabeth.schnell@kuleuven.be

Competing interest: The authors declare that no competing interests exist.

## eLife assessment

Schnell et al. report **important** differences between the strategies used by rodents and humans when discriminating different visual objects. The evidence supporting these findings is **convincing**, showing that rat performance was influenced far more by low-level cues compared to humans. It is, however, unclear to what extent these differences can be explained by the lower visual acuity of rats. This work will be of general interest to vision and cognition researchers, particularly those studying object vision.

## Introduction

Humans show high proficiency in invariant object recognition, the ability to recognize the same objects from different viewpoints or in different scenes. This ability is supported by the ventral visual stream, the so-called *what stream* (**Logothetis and Sheinberg, 1996**). A question that is repeatedly addressed in vision studies is whether and how we can model this stream by means of animal models or computational models to further examine and quantify the representations along the ventral visual stream. Computationally, researchers have recently modelled this stream by using convolutional deep neural networks (CNNs), as, for example, done by **Avberšek et al., 2021**, **Cadieu et al., 2014**, **Duyck**

et al., 2021, Güçlü and van Gerven, 2015, Kalfas et al., 2018, Kar et al., 2019, Kubilius et al., 2016, Pospisil et al., 2018, and Vinken and Op de Beeck, 2021. Lately, the rodent model has become an important animal model in vision studies, motivated by the applicability of molecular and genetic tools rather than by the visual capabilities of rodents. Past studies have examined behavioural (Alemi-Neissi et al., 2013; De Keyser et al., 2015; Djurdjevic et al., 2018; Schnell et al., 2019; Tafazoli et al., 2012; Vermaercke and Op de Beeck, 2012; Vinken et al., 2014; Zoccolan, 2015; for a review, see Zoccolan, 2015) as well as neural (Matteucci et al., 2019; Tafazoli et al., 2017; Vermaercke et al., 2014; Vinken et al., 2016) data of rodents (rats and mice) performing in visual pattern recognition tasks. The behavioural findings suggested that rats are capable of learning complex visual discrimination tasks. Here we plan to integrate computational and animal modelling approaches by using data about information processing in artificial neural networks when designing the animal experiments.

One aspect that almost all rodent studies have in common is that the exact task and stimuli are chosen based on what we know from human and monkey studies. Earlier research showed that the intuition of researchers about the complexity of visual tasks can be misleading (Vinken and Op de Beeck, 2021). Through computational CNN modelling of the tasks from previous studies, they showed that behavioural strategies that seem complex at first hand might be best modelled through relatively

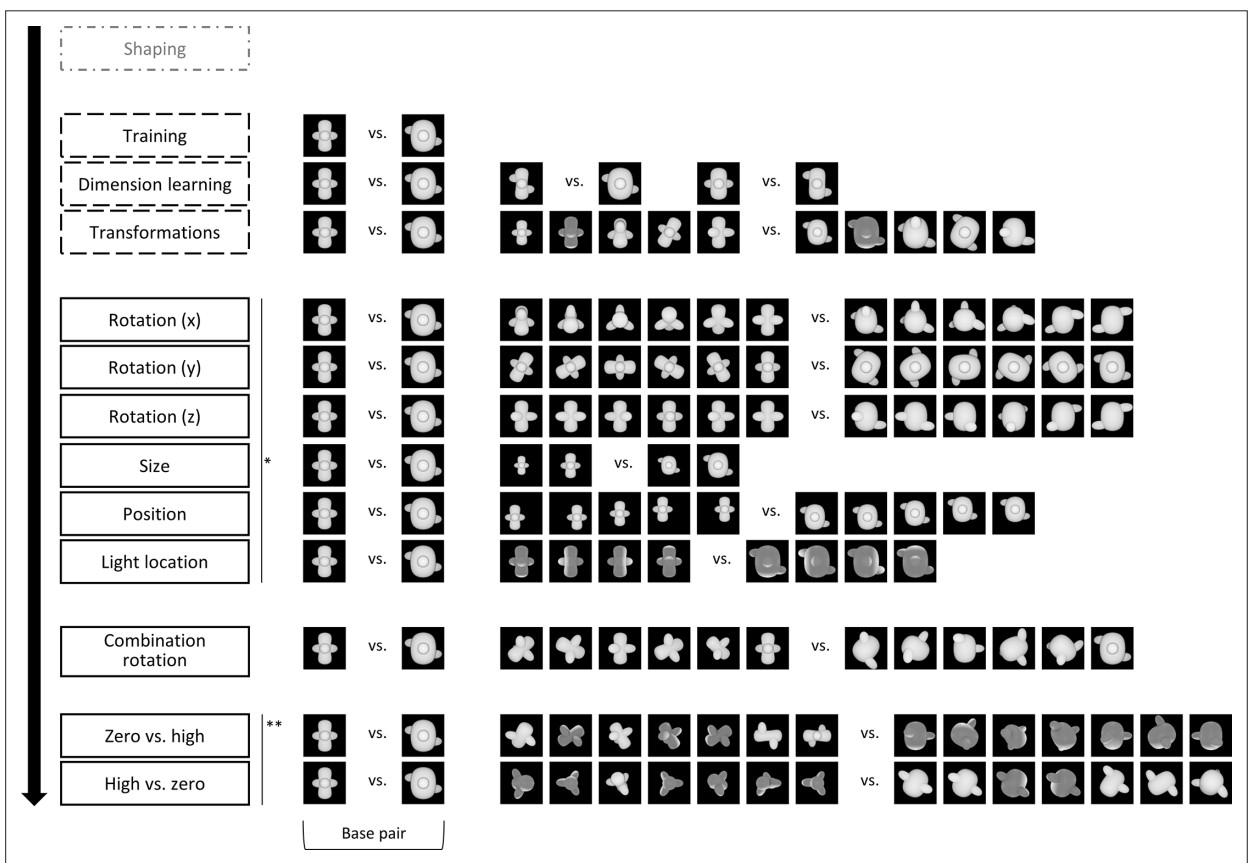

**Figure 1.** The design of the animal study, including the stimuli. Animals started with a standardized shaping procedure, followed by three training protocols, as indicated by the dashed outline. In these protocols, animals received real reward, that is, reward for touching the target. The target corresponds to the concave object in all training protocols. The rats received correction trials for incorrect answers, that is, touching the convex object. After the three training protocols, the animals went through a number of testing protocols. The order of the first six protocols (*) and the last two testing protocols (**) was counterbalanced between the animals. During testing protocols, animals received one-third old trials and two-third new trials. In the new trials, they received random reward in 80% of the trials, whereas in the old trials, they received real reward and correction trials if necessary. Again, the target in the testing protocols correspond to the concave objects, whereas the distractors correspond to the convex objects.

The online version of this article includes the following figure supplement(s) for figure 1:

**Figure supplement 1.** Design of the online human study.

**Figure supplement 2.** Trial and experimental setup information.

**Figure supplement 3.** Design of the pilot study.

early levels of processing in CNNs. They recommended that future studies could obtain more direct information about the complexity of visual tasks and behavioural strategies by incorporating neural network models in the design phase of the experiment. One way of implementing this is to train rodents in a challenging and multidimensional visual task and use CNNs to select stimulus examples targeting strategies with different levels of complexity.

In this study, we implemented this approach and created a large stimulus set that can be used for a variety of visual experiments. We decided to create the stimuli in a way that they are adaptable to different types of tasks, such as a 'simple' discrimination task or non-linear tasks (e.g. *Bossens and Op de Beeck, 2016*). We then took a subset of these stimuli and performed a visual discrimination experiment in rats (see *Figure 1* for the design). The task itself was defined in a stimulus space with two dimensions, here referred to as concavity and alignment. The stimuli consisted of a base shape that varied in concavity, with three spheres attached to it that were either horizontally aligned or misaligned. The task was then further complicated by transforming the stimuli along several dimensions that preserve the identity of the object. We started by training the animals in a base stimulus pair, with the target being the concave object with horizontally aligned spheres. Once the animals were trained in this base stimulus pair, we used the identity-preserving transformations to test for generalization. After a number of transformation phases, we selected a final stimulus set by choosing a combination of transformations based on the outcomes of a trained CNN. Using the neural network as a (basic) model for the different stages of ventral visual stream processing, we chose stimulus pairs that require either higher or lower levels of processing and thus allow us to maximally differentiate between the task strategies used by the animals. As a final part of this study, we performed an online human experiment with the same stimuli and design as the experiment for the rats, providing us with a rich three-way comparison of rat behavioural data with human behavioural data and with CNN data.

## Results

In this study, we trained and tested 11 rats and 45 humans on a complex two-dimensional discrimination task (see *Figure 1* for the design of the rat study and *Figure 1—figure supplement 1* for the design of the human study). Rats and humans were first trained in a base pair. Next, we tested their

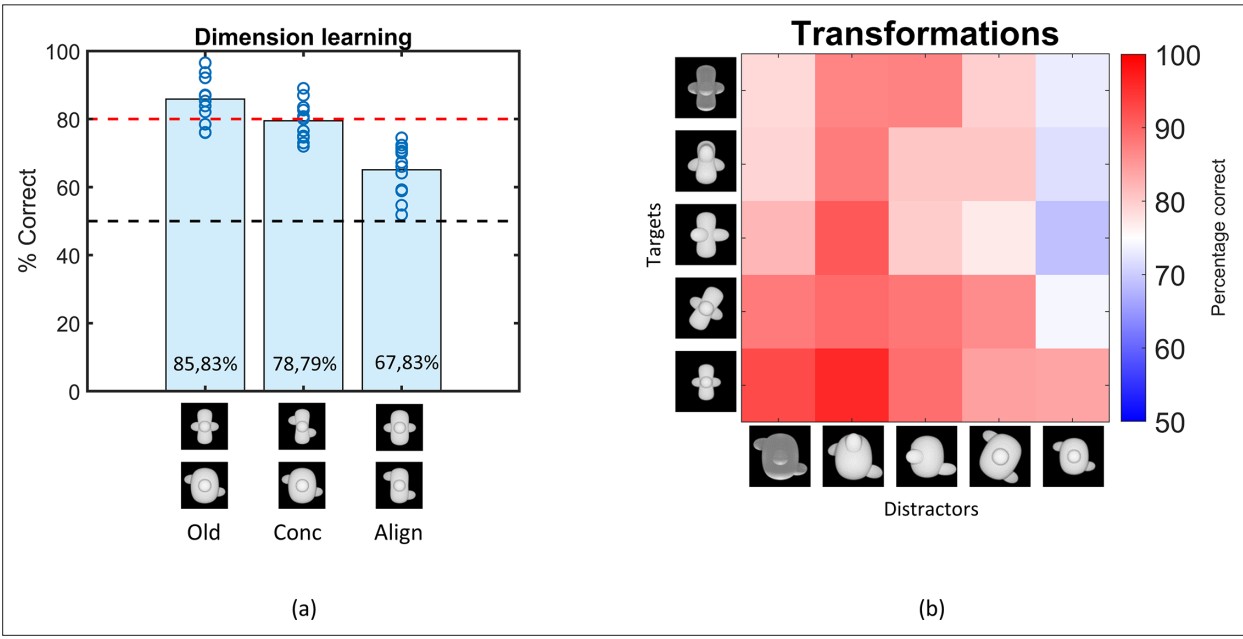

**Figure 2.** Results of the Dimension learning and Transformations training protocol. (**a**) Results of the dimension learning training protocol. The black dashed horizontal line indicates chance-level performance and the red dashed line represents the 80% performance threshold. The blue circles on top of each bar represent individual rat performances. The three bars represent the average performance of all animals on the old pair (Old), the pair that differs only in concavity (Conc), and on the pair that differs only in alignment (Align). (**b**) Results of the transformations training protocol. Each cell of the matrix indicates the average performance per stimulus pair, pooled over all animals. The columns represent the distractors, whereas the rows separate the targets. The colour bar indicates the performance correct. Testing across transformations.

ability to generalize across several image transformations. In the last two protocols of the design, we used a computational approach to select stimuli that require different visual strategies.

## Animal study

### Training

We first checked the variation in performance across phases and stimulus pairs during training. In the first *training phase*, animals were trained in the base stimulus pair, which were the maximally different target and distractor in a concavity × alignment stimulus space where each dimension was varied with four values (4 × 4 space). This training was successful for all 12 animals and lasted on average for 8.62 sessions (SD = 1.61). Animals were trained until they reached 80% performance for two consecutive sessions.

Once the animals were successfully trained, we examined whether they use both dimensions (concavity and alignment) by presenting them with two additional stimuli pairs where the target and distractor differ in only one dimension (see *Figure 2*, *dimension learning*). Performance on the old pair was similar to training performance (85.83%). The animals performed well with the stimuli that differ only along the concavity dimension (78.79%), although it was significantly lower than the performance on the base pair (paired *t*-test on rat performance, $t(11) = 3.77$, p=0.003). Performance dropped to 67.83% for the alignment-only pair, yet also significantly higher than chance level (one-sample *t*-test, p<0.0001). Overall, the *dimension learning* protocol provides evidence that the animals have picked up each of the two dimensions. This finding already excludes trivial explanations in terms of simple visual dimensions. For example, while concavity is correlated with horizontal size (distractor wider) and with overall brightness (distractor brighter, thus the opposite relevance as in the shaping phase), these simple dimensions cannot explain above-chance performance on the alignment dimension.

The third training protocol consisted of a number of small transformations, as visualized in *Figure 1* (*transformations*). Rats learned these transformations very well, with an average performance of 83.05% (see *Figure 2*). The pairwise percentage matrix in *Figure 2* shows that the distractor with the size transformation (rightmost column in the matrix) affected the rat performance the most.

The variation in performance across targets and distractors can be due to a variety of factors. This can include simple dimensions such as brightness. In the base pair, the distractor is brighter than the target. While this is the opposite from the shaping task of detecting a shape versus a black screen, visual inspection of *Figure 2* suggests that the animals perform poorer on trials in which the distractor display is not so much brighter (e.g. when it is small). To quantify this effect of brightness, we calculated the correlation between the performances in the matrix and the difference in pixel values (and thus brightness) of the stimulus pairs. This resulted in a (Pearson) correlation of –0.59 (p<0.01), suggesting that there is indeed an effect of brightness. Yet, brightness is at best a partial explanation because all percentages in the matrix are above chance, with the lowest percentage in the matrix being 68.83%, even though in some pairs the difference in pixel values is abolished or even opposite from the base pair.

Overall, the findings from the training phase and the above-chance performance on a variety of dimensions and transformations suggest that the rats have learned a pattern classification task with a level of complexity that might be competitive with other tasks in the rodent literature.

The six protocols that test generalization to various transformations with new, untrained images are associated with performances lower than 80% (binomial test, see *Supplementary file 1a* [lower table] for detailed table with results), but significantly higher than chance level (see *Supplementary file 1a* [lower table]). The pairwise percentage matrices of the animals in *Figure 3* provide a more detailed view of what is happening in every test, and *Figure 3—figure supplement 1* shows the individual accuracy for each animal. The distractor has a higher impact on performance than the target in some tests. *Supplementary file 1b* shows the marginal means and standard deviation for each target and distractor for these two test protocols. From these means it is clear that there is a higher variation in the performance between distractors in *rotation X* (52–65%) and *rotation Z* (56–73%) than between targets (55–60% resp. 60–66%). The same happens in the size test protocol.

After these first six test protocols, the animals were presented with a schedule where all three rotations are combined (see *Figure 1*). On the new stimuli, the animals performed 58.56%, which is rather low, but still significantly different from chance level (binomial test on pooled performance of all animals: p<0.0001; 95% CI [0.57;0.60]).

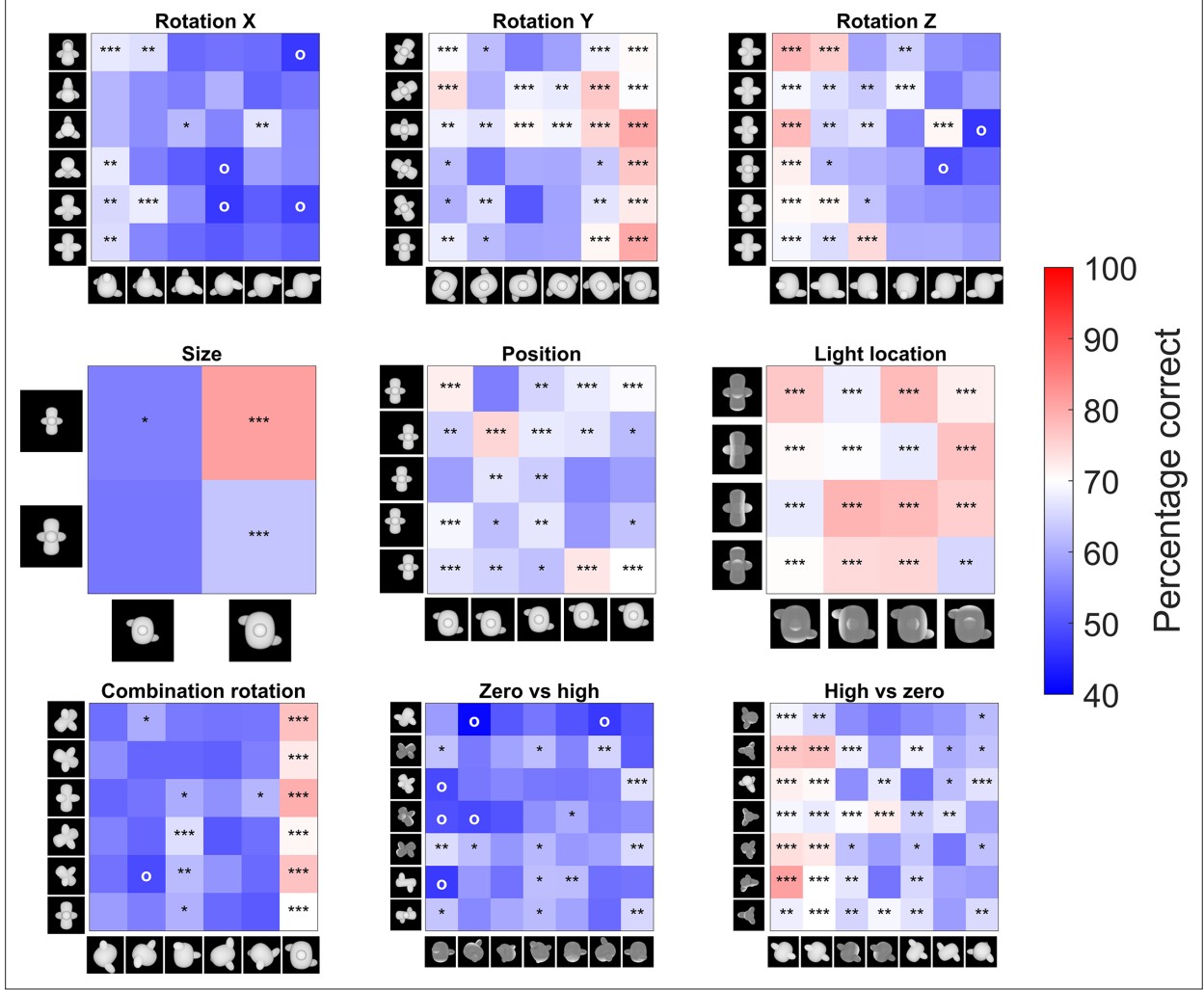

**Figure 3.** Pairwise percentage matrices of all nine testing protocols for the rat data. The colour bar indicates the percentage correct of the pooled responses of all animals together. The redder a cell is, the higher the average performance. Values below 40% accuracy are indicated in the highest intensity of blue. Cells with an 'o' marker indicate a below-chance performance, whereas cells with an *, **, or *** marker indicate a performance that is significantly higher than chance level (p-value <0.05, <0.01, or <0.001 respectively). This was calculated with a binomial test on pooled performance of all animals.

The online version of this article includes the following figure supplement(s) for figure 3:

**Figure supplement 1.** Individual rat accuracy for each testing protocol.

### Testing computational levels of complexity

For the final two test protocols, we used a CNN to find image pairs that would contrast strategies based upon a different stage in visual processing, with either early layers having lower performance than high layers (*zero* vs. *high*), or early layers having better performance than high layers (*high* vs. *zero*). Rat performance was particularly low for *zero* vs. *high* (56.47%), yet still significantly different from chance level when averaged across all stimulus pairs (binomial test on pooled performance of all animals; p<0.0001; 95% CI [0.55;0.58]). In contrast, rats were able to solve the *high* vs. *zero* pairs not only better than chance (average: 64.84%; binomial test on pooled performance of all animals; p<0.0001; 95% CI [0.63;0.66]), but also significantly better than *zero* vs. *high* (paired *t*-test on rat performance, $t(10) = -4.49$, p=0.0012). This suggests that rats align with lower levels of processing when we purposely select image pairs that are optimized to contrast different levels of the visual processing hierarchy.

Next we checked how much individual CNN layers can predict the variation in behavioural performance across image pairs when we take all test protocols together. We calculated the correlation of

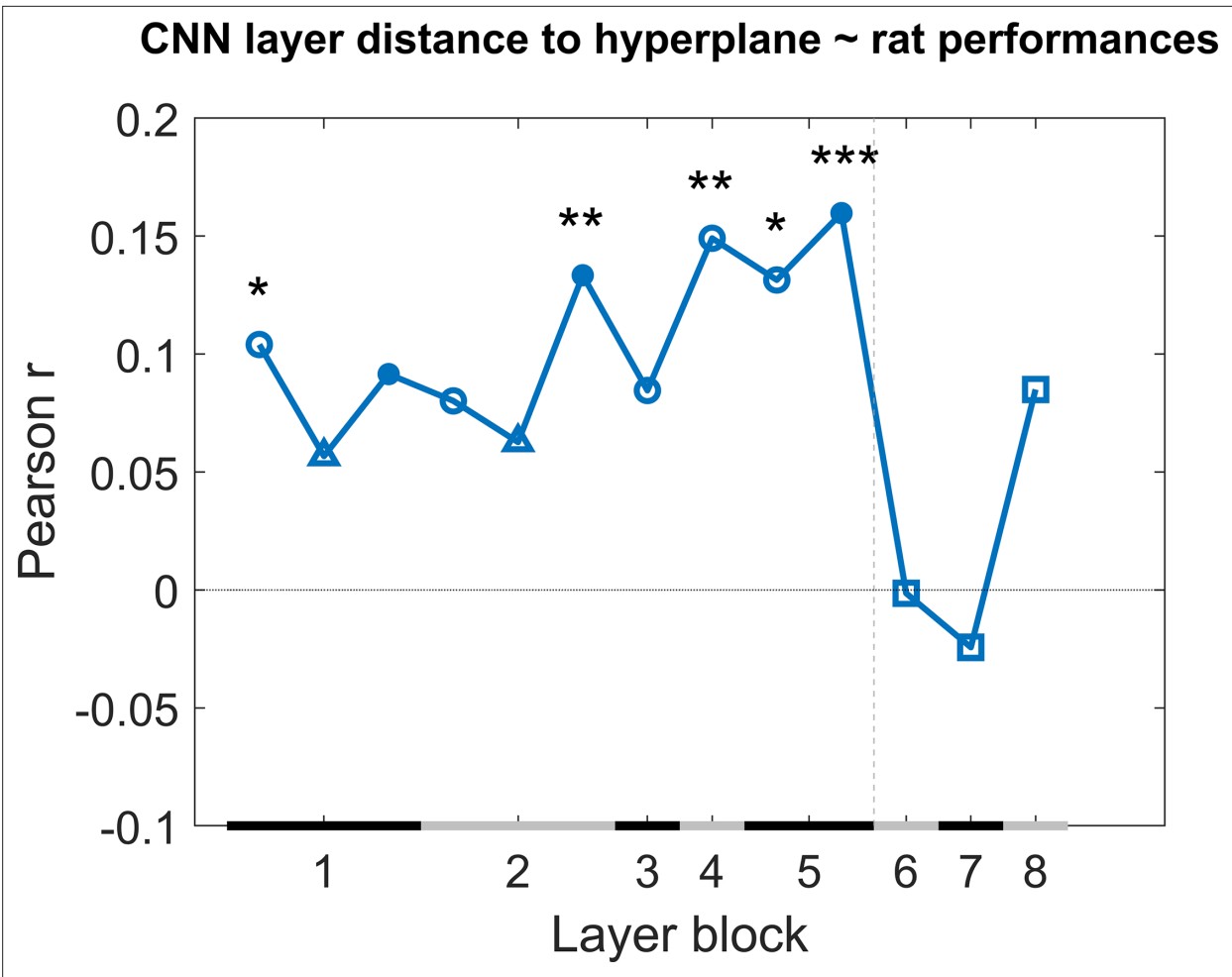

**Figure 4.** Correlation of the classification score for single target/distractor pairs between single convolutional deep neural network (CNN) layers and the rat performance for all nine test protocols together. The black and grey horizontal lines on the x-axis indicate the layer blocks (block 1 consisting of conv1, norm1, pool1; block 2 consisting of conv2, norm2, pool2; block 3–4 corresponding to conv3-4, respectively; block 5 consisting of conv5, pool5; block 6-7-8 corresponding to fc6-7-8, respectively). The vertical grey dashed line indicates the division between convolutional and fully connected layer blocks. The horizontal dashed line indicates a correlation of 0. The different markers indicate different sorts of layers: circle for convolutional layers, triangle for normalization layers, point for pool layers, and squares for fully connected layers. The asterisks indicate significant correlations according to a permutation test (*<0.05, **<0.01, and ***<0.001).

the generalization across image pairs between the CNN classifier (summarized in Figure 8) and the rat performance of all nine test protocols. This correlation includes a total of 287 image pairs, that is, all image pairs of all nine test protocols together. We did this by concatenating all performances of the animals into one array and all classification scores of the network into another array, and calculating the correlation between these two arrays to retrieve a correlation for each network layer. The results are displayed in *Figure 4*. Overall, we see quite low correlations, but several convolutional layers nevertheless show a significant positive correlation (permutation test) with the behavioural pattern of performance at the image pair level.

Even though some of the correlations are significant, they are low. This could indicate that no CNN layer is able to capture what rats do. Alternatively, it could be caused by a very low reliability of the behavioural data. To test the reliability of the variations in behavioural performance between stimulus pairs in all nine test protocols, we calculated the split-half reliability, as previously done in *Schnell et al., 2023*, resulting in a correlation of 0.40. By applying the Spearman–Brown correction, we obtain a full-set reliability correlation of 0.58. This correlation is much higher than the correlations with individual CNN layers.

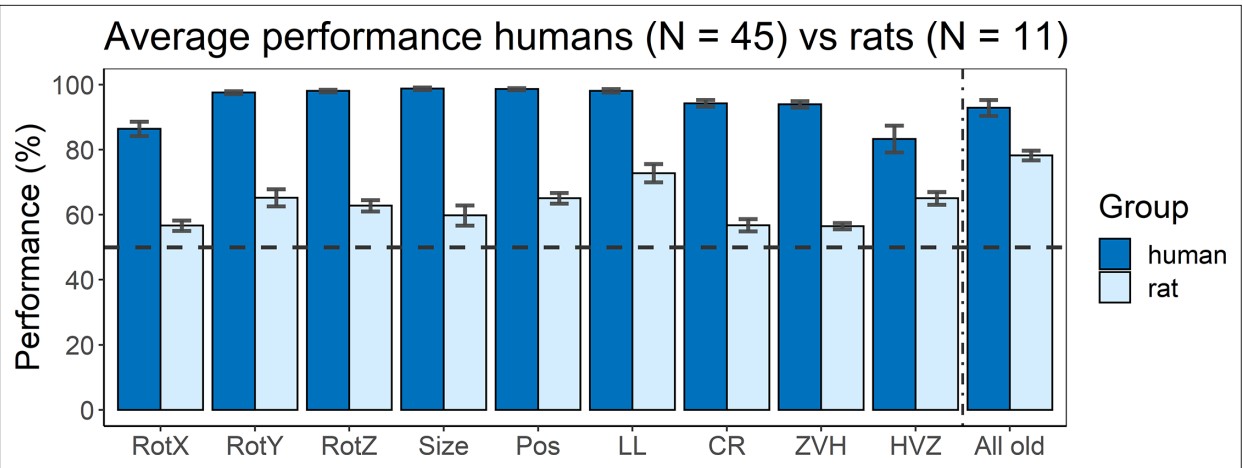

**Figure 5.** Average performance of humans versus rats. On the x-axis, the nine test protocols in addition to the performance on all old stimuli are presented in the following order: rotation X (RotX), rotation Y (RotY), rotation Z (RotZ), size, position (Pos), light location (LL), combination rotation (CR), zero vs. high (ZVH), high vs. zero (HVZ), and all old. The dashed horizontal line indicates chance level. The error bars indicate standard error over humans/rats (N = 45 for humans, N = 11 for rats).

It is possible that rat performance would be based upon multiple levels of processing, in which case we would need a combination of layers in order to explain the variation in performance across stimulus pairs. Given the low correlation between neighbouring layers (**Supplementary file 1c**), a multiple linear regression was calculated with the classification scores of the 13 layers as 13 regressors, and the rat performances as response vector. The results of this regression indicate a significant effect of the classification scores ($F(287,273) = 2.22$, p=0.00907, $R^2 = 0.10$). Further investigating the 13 predictors showed that the later convolutional layers 8–10 of the network were significant predictors in the regression model (see **Supplementary file 1d** for results of the regression model). The $R^2 = 10$ of the full model would correspond to a correlation of around 0.32. This is better than the correlation of single layers, but still clearly smaller than the reliability of the rat data of 0.58. In conclusion, the CNN model provides a partial explanation of how the performance of rats varies across image pairs.

Given the relevance of convolutional layers, we can expect that relatively basic visual features might partially explain the behavioural strategy of rats. This includes dimensions such as brightness and pixel-based similarity. To get a first indication of the relevance of these features, we calculated the correlation across image pairs between rat performance and brightness and pixel similarity. Here we found a correlation of 0.34 for pixel similarity and 0.39 for brightness, suggesting that these two visual features partially explain our results when compared to the full-set reliability of rat performance (0.58).

## Human study

A final part of this study was to include an online human study that follows the same design as the animal part. *Figure 5* shows the average performance of humans (dark blue) versus rats (light blue) for all nine test protocols, as well as their performance on the old stimuli that were added in (or during) the testing protocols as quality control. Overall, humans performed better on all tests protocols than rats, with an average performance over all tests of 94.34% (humans) and 62.29% (rats). There was already a difference in terms of training performance (humans 92.86% vs. rats 77.84%), but the difference on the test protocols is larger. We subtracted the training performance of humans or rats from the testing performance of humans or rats, respectively, and even with this normalization for training performance there is still a significantly higher test performance in humans compared to rats ($t(16) = -6.47$, p<0.0001). Thus, not surprisingly, the degree of invariance in this object classification task is higher for humans compared to rat.

The variation in performance across test protocols and across image pairs can give an indication of the strategies that each species follows. Overall, humans and rats show a mild correspondence in terms of which image pairs are more difficult, with a human–rat correlation of 0.18 across all image pairs of the nine test protocols (p<0.001 with permutation test). Albeit significant, this correlation is clearly lower than the maximum value that could be obtained given the reliability of the data. The

split-half reliability of the human data was 0.46, corresponding to a full-set reliability of 0.63. We reported above that full-set reliability is 0.58 for the rat data, resulting in a combined reliability of 0.60 (calculated as described in *Op de Beeck et al., 2008*). Thus, after taking data reliability into account there remains a pronounced discrepancy between rats and humans in terms of how performance varies across image pairs.

The main question of this study is how this discrepancy relates to computationally informed strategies. If we take a closer look specifically at the two CNN-informed test protocols (*zero* vs. *high* and *high* vs. *zero*), we see an opposite behaviour between animals and humans. Humans performed significantly better in the *zero* vs. *high* protocol, that is, where we used stimuli where the earlier layers of the network perform worse than the higher layers, than in the *high* vs. *zero* protocol (paired *t*-test: *t*(44) = 2.85, p=0.0067). Rats, however, show the opposite (see above for statistics). There even is a significant interaction between species and test protocol (unpaired *t*-test: *t*(54) = 2.50, p=0.016). This suggests a different strategy between animals and humans: rats use strategies that are captured in the lower layers of the network, and thus correspond more to low-level visual processing. Humans, however, tend to rely more on strategies captured by the higher layers of the network, and thus we are looking at more high-level visual processing.

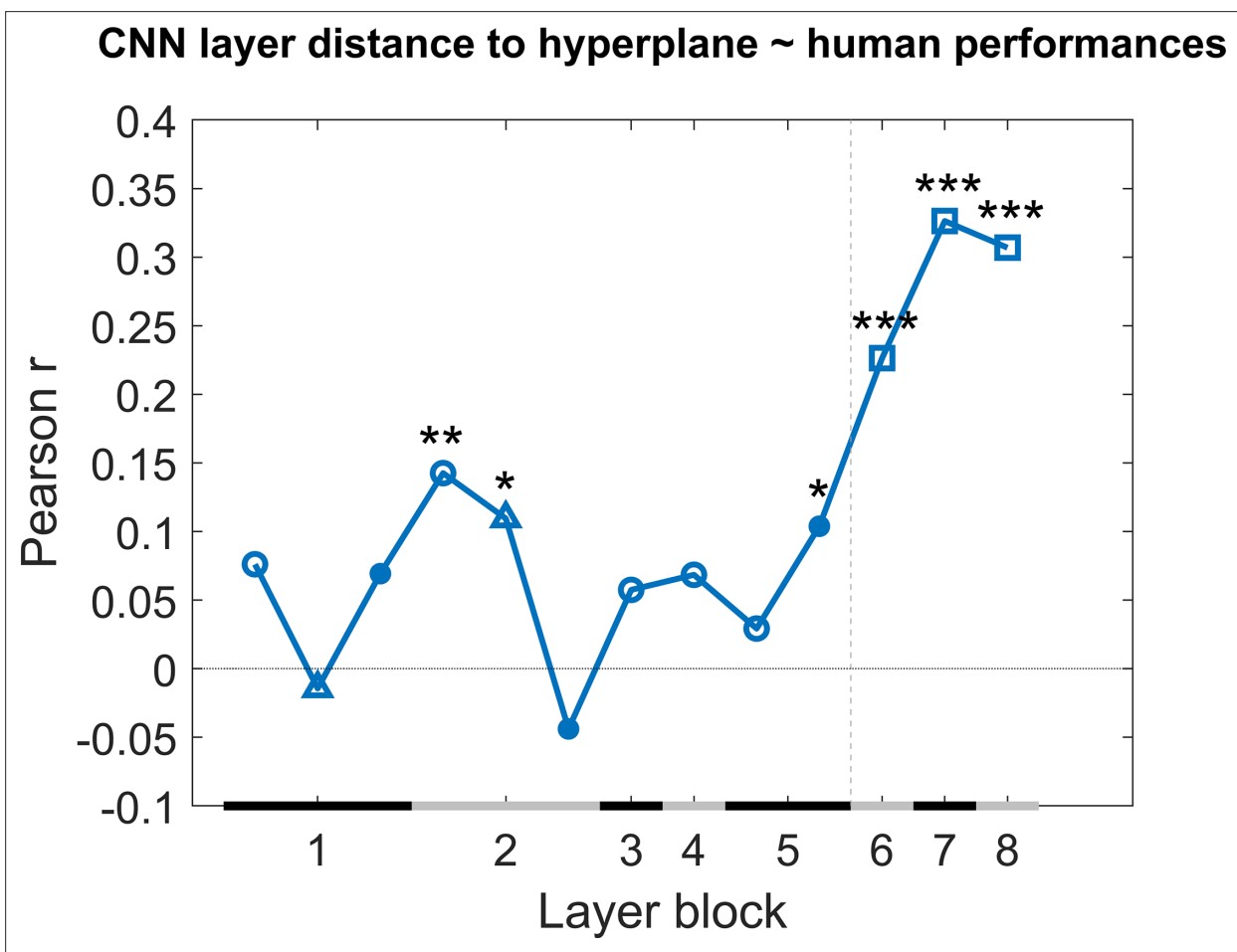

**Figure 6.** Correlation of the classification score for single target/distractor pairs between single convolutional deep neural network (CNN) layers and the human performance for all nine test protocols together. The naming convention on the x-axis corresponds to the layers of the network, identical as in *Figure 4*. The black and grey horizontal lines on the x-axis indicate the layer blocks (block 1 consisting of conv1, norm1, pool1; block 2 consisting of conv2, norm2, pool2; block 3–4 corresponding to conv3-4, respectively; block 5 consisting of conv5, pool5; block 6-7-8 corresponding to fc6-7-8, respectively). The vertical grey dashed line indicates the division between convolutional and fully connected layer blocks. The horizontal dashed line indicates a correlation of 0. The different markers indicate different sorts of layers: circle for convolutional layers, triangle for normalization layers, point for pool layers, and squares for fully connected layers. The asterisks indicate significant correlations according to a permutation test (*<0.05, **<0.01, and ***<0.001).

As a next step, we calculated the correlation between the generalization across image pairs between the CNN classifier and the human performance of all nine test protocols in an identical manner as for the rat performance (*Figure 4*). The results are displayed in *Figure 6*. Overall, we see quite high correlations, especially in the higher layers. This pattern across layers is very different from the pattern in rats where the highest layers showed no correlations, which again suggests that, despite successful generalization, rats rely on decisively lower-level strategies than humans in the same discrimination task.

A multiple linear regression was calculated in an identical manner as we did with the rat performance. The results of this regression indicate a significant effect of the classification scores ($F$(287,273) = 6.8, p<0.0001, $R^2$ = 0.25). Further investigating the 13 predictors showed that in particular the fully connected layers 11–13 of the network were strong predictors in the regression model (see for results *Supplementary file 1e* of the regression model).

Given the high correlations with fully connected layers, we would expect to not find much evidence for an influence of basic visual dimensions such as brightness and pixel-based similarity. Indeed, we find small correlations with variation in human behavioural performance across image pairs for pixel similarity (0.12) and brightness (–0.12).

## Discussion

In this study, we trained and tested rats and humans in a discrimination task using two-dimensional stimuli, with the two dimensions being concavity and alignment. We tested generalization across a range of viewing conditions. For the last two testing protocols, we used a computational approach to select the stimuli in terms of specifically dissociating low and high stages of processing. Rats were able to learn both dimensions (concavity and alignment) and showed a preference for concavity. Their performance on the testing protocols revealed a wide variety in percentage correct: for some test protocols, they performed just above chance level, for example, *zero* vs. *high*, whereas for others they could easily reach about 70% correct (*position*). Humans, on the other hand, performed better overall, with performances of 80% or higher on the testing protocols. Addressing the question of the complexity of the underlying strategies, rats performed best on the test protocol designed to specifically target lower levels of processing, whereas humans performed best on the high-level processing protocol. Likewise, direct comparisons with artificial neural network layers showed that the variation of rat performance across images was best explained by late convolutional layers, whereas human performance was most associated with representations in fully connected layers.

All animals started by being trained in three training protocols. The first *training* protocol only included one image pair, the base pair, containing the most different target and distractor without any further transformations. Learning of the individual dimensions of concavity and alignment was investigated through the *dimension learning* protocol. The results from this *dimension learning* protocol indicate that our rats have more difficulties learning the alignment dimension as opposed to the concavity dimension. One possible explanation for the superior performance on the concavity dimension could be that the animals were partially solving the task such that the brighter stimulus, that is, the convex base shape, is the distractor and that their strategy is to pick the stimulus with the lowest brightness. This was confirmed by analyses on the third training protocol (*transformations*) that included small transformations along various dimensions. Nevertheless, the rats still performed above-chance level for trials in which the brightness differences were reversed, indicating that other dimensions are involved and overrule a contribution from brightness. Similar findings have been obtained in human behaviour and neuroscience. For example, despite the clear category selectivity in regions such as the fusiform face area, the selectivity in these regions is also modulated very strongly by various low-level dimensions (*Yue et al., 2011*). With regard to the size and position transformations, it is important to keep in mind that the animals were freely moving in the touchscreen chambers, and so even for the original base pair were already undergoing changes in retinal size and retinal position. What we manipulate is rather the size and position relative to the rest of the set-up (e.g. relative to screen position and size).

After these three training protocols, the animals were tested for generalization in a variety of testing protocols, each testing a separate transformation on the stimuli. The first six test protocols included rotation along all the three axes, size, position, and light location, followed by a test protocol in which we combined the rotation along the three axes. Overall, we found that the performance of

the animals on these test protocols is affected by these transformations, but still significantly above chance in each protocol. Studies in the literature would often stop here or proceed by systematically testing even larger transformations. Stimulus choices are based upon intuitions of what strategy animals might be using and upon theories of how visual perception works. However, in some cases, a further computational modelling of the task and stimuli finds that what intuitively seems like a task of a particular complexity might not be so complex after all. The first tests of invariant object recognition seemed impressive, but were found to be easily solved with earlier layers of processing (*Minini and Jeffery, 2006*; *Vinken and Op de Beeck, 2021*). This was recently also highlighted by relatively simple pixel-based analyses (*Kell et al., 2020*). As another example, *Vinken and Op de Beeck, 2021* have used a computational approach to further investigate the levels of information processing in rodents by comparing three hallmark studies that provided evidence for higher order visual processing in rodents (*Djurdjevic et al., 2018*; *Vinken et al., 2014*; *Zoccolan et al., 2009*) with CNNs. They found that for all three studies the low- and mid-level layers captured the rat performances best, providing thus evidence against the previously concluded high-level visual processing in rodents.

For these reasons, we decided to directly test image pairs through computational modelling with CNNs and select pairs that are particularly suited for dissociating different levels of processing. Stimuli were chosen by a CNN from a very large set of possible stimuli and combinations, such that the higher layers and the lower layers of the network make distinct errors on classifying the stimuli (*zero* vs. *high* and *high* vs. *zero* protocol), and thus are diagnostic of the level of underlying visual strategies. We chose to work with Alexnet as this is a network that has been used as a benchmark in many previous studies (e.g. *Cadieu et al., 2014*; *Groen et al., 2018*; *Kalfas et al., 2018*; *Nayebi et al., 2023*; *Zeman et al., 2020*), including studies that used more complex stimuli than the stimulus space in this study. The stimuli of the *zero* vs. *high* protocol included stimuli where the higher layers of the network performed better than the lower layers, and thus they address higher-level visual processing. The opposite can be said for the *high* vs. *zero* protocol, which includes stimuli that specifically target lower-level visual processing, given that the lower layers of the network perform best on these stimuli. After presenting these stimuli to the animals, we found that our rats performed best in the *high* vs. *zero* protocol, suggesting that they focus on low-level visual cues to solve this discrimination task. We found the opposite CNN pattern for humans, indicating that they use high-level visual processing. These findings provide more direct information about the level of processing that underlies the behavioural strategies compared to overall performance or to effects of image manipulations.

This is a new promising way to design experiments in a way that is computationally informed rather than based on researcher intuitions or qualitative predictions. It is in line with the literature that a typical deep neural network, AlexNet and also more complex ones, can explain human and animal behaviour to a certain extent but not fully. The explained variance might differ among CNNs, and there might be CNNs that can explain a higher proportion of rat or human behaviour. Most relevant for this study is that CNNs tend to agree in terms of how representations change from lower to higher hierarchical layers because this is the transformation that we have targeted in the zero vs. high and high vs. zero testing protocols. *Pinto et al., 2008* already revealed that a simple V1-like model can sometimes result in surprisingly good object recognition performance. This aspect of our findings is also in line with the observation of *Vinken and Op de Beeck, 2021* that the performance of rats in many previous tasks might not be indicative of highly complex representations. Nevertheless, there is still a relative difference in complexity between lower and higher levels in the hierarchy. That is what we capitalize upon with the zero vs. high and high vs. zero protocols. Thus, it might be more fruitful to explicitly contrast different levels of processing in a relative way rather than trying to pinpoint behaviour to specific levels of processing.

Partially thanks to these computationally inspired tests, our total dataset finds a marked dissociation between how humans and rats solve this object recognition task. Even in the sessions where only the old pairs are shown, the animals performed lower than humans. This was most likely due to motivation and/or distractibility. Our analyses show dissociation between humans and rats most convincingly by correlating the variation in performance across image trials with the predictions of CNN layers. There were significant correlations with multiple layers in both species. In humans, the most pronounced correlations were present for the highest, fully connected layers, while in rats correlations were limited to low and middle convolutional layers. This is the most direct evidence available in the literature that rats resolve object recognition tasks through a very different and computationally simpler strategy

compared to humans. The CNN approach does not inform us how we can verbalize this simpler strategy, but based upon earlier work (*Schnell et al., 2023*; *Vermaercke and Op de Beeck, 2012*) we would hypothesize that rats rely upon visual contrast features (e.g. this area is darker/lighter than that other area). Such contrast features are also used by humans and monkeys, for example, for face detection (*Ohayon et al., 2012*; *Sinha, 2002*), but in addition humans have access to more complex strategies that, for example, refer to complex shape features such as aspect ratio and symmetry (*Bossens and Op de Beeck, 2016*). Tests in this study reveal that other features that partially explain rat performance include basic dimensions such as brightness and pixel-based similarity, the latter being a proxy for retinotopically-based computations that are expected to be present in convolutional layers.

Our analyses of rat behaviour and DNN modelling do not take into account potential trial-to-trial variability in the distance and position of the rat's head. From earlier work we can derive that rats typically make their decision from about 12 cm from the stimulus display (*Crijns and Op de Beeck, 2019*), but we have no information on trial-to-trial variability. We can hypothesize about the possible effect. If such variability would exist, then it would artificially increase the variation in distance and position during training, and thus help the animals achieve higher levels of invariance during testing. As a consequence, the difference between rat and human performance in terms of inferred level of processing might even increase under more controlled circumstances.

For future studies, it will be highly valuable to use this computational informed strategy on a wider battery of behavioural tasks, as well as a wider range of species such as tree shrews and marmosets (*Callahan and Petry, 2000*; *Kell et al., 2020*; *Kell et al., 2021*; *Meyer et al., 2022*; *Petry et al., 2012*; *Petry and Bickford, 2019*). One step further, we can use the information from computational modelling together with behaviour and how it differs among stimuli to further select stimuli for neurophysiological investigations of neuronal response properties along the visual information processing hierarchy, in this way following experimental designs that are optimized for highlighting the primary differences between processing stages and between species.

## Methods
### Animal study
#### Animals

A total of 12 male outbred Long–Evans rats (Janvier Labs, Le Genest-Saint-Isle, France) started this behavioural study. Out of these 12 animals, 2 were tested extensively in a first pilot study and were included in the remainder of the study as well. All animals were 11 weeks old at the start of shaping and were housed in groups of four per cage. Each cage was enriched with a plastic toy (Bio-Serv, Flemington, NJ), paper cage enrichment, and wooden blocks. Near the end of the experiment, one animal had to be excluded because of health issues. During training and testing, the animals were food restricted to maintain a body weight between 85 and 90% of their underprived body weight. They received water ad libitum. All experiments and procedures involving living animals were approved by the Ethical Committee of the University of Leuven and were in accordance with the European Commission Directive of September 22, 2010 (2010/63/EU).

#### Setup

The setup is identical to the one used by *Schnell et al., 2019* and *Schnell et al., 2023*. A short description will follow here. The animals were trained and tested in four automated touchscreen rat-testing chambers (Campden Instruments, Ltd., Leicester, UK) with ABET II controller software (v2.18, WhiskerServer v4.5.0). The animals performed one session per day and each session lasted for 100 trials or 60 min, whichever came first. A reward tray in which sugar pellets (45 mg sucrose pellets, Test-Diet, St. Louis, MO) could be delivered was installed on one side of the chamber. On the other side of the chamber, an infrared touchscreen monitor was installed. This monitor was covered with a black Perspex mask containing two square response windows (10.0 × 10.0 cm). A shelf (5.4 cm wide) was installed onto this black mask (16.5 cm above the floor) to force the animals to attend to the stimuli and view the stimuli within their central visual fields. Close proximity to the screen was enough to elicit a response because the screens are infrared. As the position of the rats in the touchscreen setup is not fixed, the actual size and position of the stimuli might vary in retinal coordinates. In a previous study, we manipulated the cycles per degree of stimuli in an orientation discrimination task and estimated

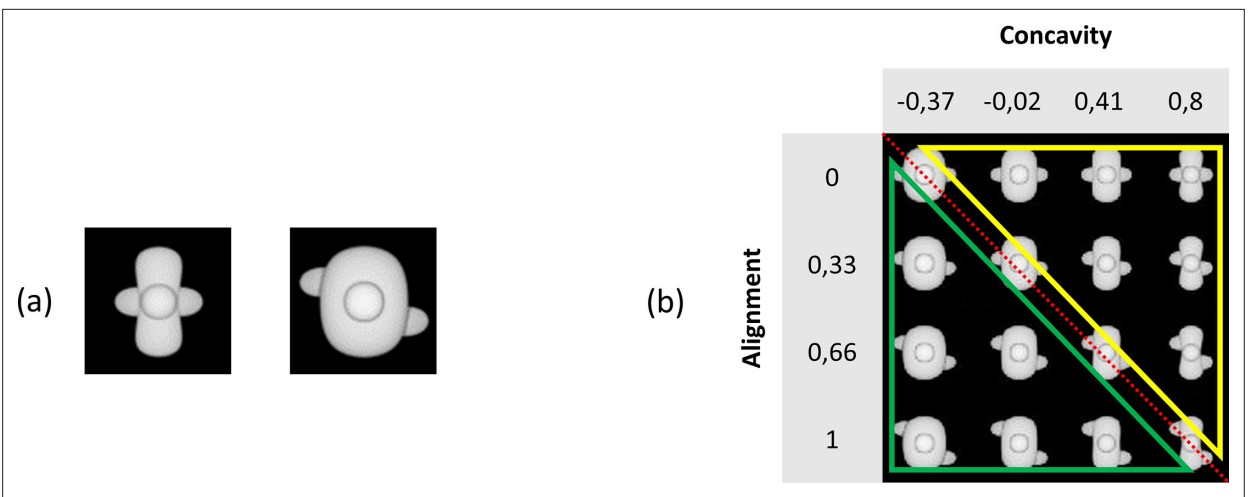

**Figure 7.** Illustration of the base pair and our stimulus grid. (**a**) The base pair of the main experiment. (**b**) The chosen 4 × 4 stimulus grid. The red diagonal dotted line indicates the ambiguous stimuli that can be seen as target as well as distractor. All stimuli below this line (green triangle) indicate the distractor sub-grid, whereas all stimuli above this line (yellow triangle) highlight the target sub-grid.

The online version of this article includes the following figure supplement(s) for figure 7:

**Figure supplement 1.** Illustration of the 4 × 11 stimulus grid.

**Figure supplement 2.** The pixel dissimilarity matrix of the 4 × 11 stimulus grid.

**Figure supplement 3.** Identity-preserving transformations on one of the basic stimuli.

that the decision distance of rats in this setup lies around 12.5 cm from the screen (*Crijns and Op de Beeck, 2019*). *Figure 1—figure supplement 2* shows the timeline graphic of a correct and incorrect trial as well as images of the experimental setup.

## Stimuli

Stimuli were created using the Python scripting implementation of the 3D modelling software Blender 3D (version 2.93.3) and measured 100 × 100 pixel. In general, the stimuli were objects that consisted of a body (base) with three spheres attached to it. A first step was to alter two dimensions of the object, namely, the concavity of the base and the alignment of the three spheres. The base was made either concave or convex by increasing (convex) or decreasing (concave) the base parameter. The alignment of the spheres was altered by changing the placement of the left and the right spheres. These spheres could either be horizontally aligned or misaligned. In the misaligned case, the spheres were placed diagonally from upper left to lower right. *Figure 7a* shows two example stimuli, the ones that later were selected as the so-called 'base pair'. Next, additional exemplars were created by uniformly tiling the two-dimensional stimulus space between these two example stimuli. We decided to create 11 levels of the concavity dimension and 4 levels of alignment. This already yields 44 stimuli (see *Figure 7—figure supplement 1*). We chose these levels of concavity and alignment based on the pixel dissimilarity of the stimuli (see *Figure 7—figure supplement 2*). The final goal was to construct a 4 × 4 stimulus grid (*Figure 7b*) by selecting a subset of the 4 × 11 stimulus grid. We chose a large number of concavity levels as this ensures flexibility in the calibration of the two dimensions relative to each other.

We added identity-preserving transformations to the stimuli, such as rotation among the x-axis, y-axis, and z-axis in six different angles (0°–180° in steps of 30°), as well as changing the light location (left, under, up, right, front) and finally the size and position. The latter two transformations were implemented using Python (3.7.3). Excluding the size and position transformation, these transformations resulted in a total set of 75,460 stimuli (4 [alignment] * 11 [concavity] * 7 [x-axis rotation] * 7 [y-axis rotation] * 7 [z-axis rotation] * 5 [light location] = 75,460 stimuli). *Figure 7—figure supplement 3* shows examples of these transformations, and *Figure 1* shows an overview of all image pairs that were used in this study.

## Protocols

Once the pilot was finished (see *Supplementary file 2* for details), we set up the experiment and chose our stimuli. We started by reducing the 4 × 11 stimulus grid to a 4 × 4 stimulus grid (see *Figure 7b*). All stimuli on the diagonal can be seen as ambiguous stimuli (four stimuli in total) as they can be identified as a target as well as a distractor. The six stimuli above this diagonal create the target part of the grid, and the six stimuli below this diagonal resemble the distractor sub-grid.

The different phases of the experiment are shown in *Figure 1*, and this figure shows all stimuli that were used. In the main training phase, we trained the animals in the maximally different stimuli that are placed at the very ends of the corners (*Figure 7a*). We refer to this as the base pair. After this training phase, the experiment consisted of two further training protocols. In the dimension learning training phase, we pushed the animals to learn both dimensions (concavity and alignment) by presenting them two additional stimuli pairs from *Figure 7b* in which the target and distractor differ in only one dimension. A third training protocol (*transformations*) consisted of stimuli with some small transformations, such as 30° rotation along the x-axis, 30° rotation along the y-axis, 30° rotation along the z-axis, light location below, and size reduction of 80%, resulting in a total of 25 possible stimulus pairs (every combination of target–distractor with the five transformed stimuli). During these two training protocols, one-third of the trials were so-called 'old trials' with the base pair. Correction trials were given if an animal answered incorrectly, that is, the same trial was repeated until the animal answered correctly. These correction trials were excluded from the analyses. In all trials, rats received a reward for touching the correct screen, that is, the screen with the target.

After these three training protocols, the testing part of the experiment included nine test protocols. The crucial defining difference between these test protocols and the prior training protocols is that rats received a reward randomly in 80% of the trials with new stimulus pairs, and no correction trials were given for an incorrect response. This random reward is important to keep the animals motivated during the testing protocols and measure real generalization, and not training behaviour. We have used a similar approach in the past, where we rewarded the animals in every testing trial (*Schnell et al., 2019*; *Vinken et al., 2014*). One-third of the trials in all test protocols consisted of old trials with the base pair, and here, the animals received reward for touching the target and correction trials were shown if necessary. Regularly, we inserted a *dimension learning* session in between two test sessions to maintain the performance high enough on training stimuli, especially for the animals in which we saw a drop in performance on the base pair. We excluded any test sessions where the performance on the base pair stimuli dropped to below 65% and the performance on this base pair was not included in the accuracy calculations.

The first six test protocols included one protocol for each transformation, that is, *rotation X, rotation Y, rotation Z, light location, size,* and *position*. The order in which these first six test protocols were given to the animals was counterbalanced between the animals. The stimuli that were used in these six test protocols can be seen in *Figure 1*, and every combination of target–distractor per test protocol was presented to the animals. For the rotation protocols, we used rotation degrees in steps of 30°, ranging from 30° to 180°. This resulted in 36 possible stimulus pairs for each of the three rotation protocols. In the *light location* protocol, we used stimuli where the light location was set at four different positions (below, left, right, and up), resulting in 16 possible stimulus pairs for this protocol. In the *size* protocol, we selected targets and distractors that were 80 and 60% reduced in size compared to the original, training pair. This protocol included four possible stimulus pairs. And finally, in the *position* protocol, we changed the position of the 80% reduced in size stimuli and placed the objects in the lower-left corner, lower-right corner, centre, upper-left corner, and upper-right corner. We have a total of 25 possible stimulus pairs for this protocol.

After these six test protocols, we presented the animals with six targets and six distractors where all three rotations were combined (*combination rotation*), that is, x-, y-, and z-axis were rotated with the same degree (ranging from 30° to 180°, in steps of 30°). This resulted in a total of 36 new stimulus pairs. Again, no correction trials were included after the trials where rotated stimuli were shown and animals received random reward in 80% of the trials. One-third of the trials consisted of the stimulus pair from the first *training* phase (i.e. the base pair), and here, correction trials were given after an incorrect response and real reward was given to the animals.

In a final set of two test protocols, we created a CNN-informed stimulus set. The details of the computational modelling are explained in the next section. The first protocol (*zero* vs. *high*) included

stimuli in which the lower layers of the network performed around chance level (i.e. target–distractor difference in classification scores [difference in signed distance to hyperplane] of about 0), whereas the higher layers scored high (see section 'Computational modelling'). The second protocol (*high vs. zero*) included stimuli where the network did the opposite. That is, the earlier layers performed well, whereas the higher layers performed around chance level. The order of the two test protocols was counterbalanced between the animals. Each of these test protocols included 7 targets and 7 distractors, giving a total of 49 new stimulus pairs.

Animals stayed in each session for 60 min or until they reached 100 training trials or 120 testing trials. We used an intertrial interval (ITI) of 20 s and a timeout of 5 s during training sessions. This timeout was only used in incorrect trials. From another pilot study in the lab, we noticed we could decrease the ITI and timeout without affecting the rats' performance. Therefore, we decided to use an ITI of 15 s and timeout of 3 s during testing, and to increase the number of trials during a testing session to 120 trials. The stimuli remained on the screen until the animals made a choice and so there was no time limit for the animals.

Each protocol was run for multiple sessions per animal. Given that we were interested in how performance would vary across stimulus pairs, we completed more sessions for the protocols that included more stimulus pairs. *Supplementary file 1f* indicates the average number of trials per test protocol for all rats together.

One animal was not placed in the *transformations* phase as it was the slowest animal during training. However, its performance on the test protocols did not significantly differ from the other animals. We tested this by calculating the correlation of the variation of performance across stimulus pairs for each rat with the pooled responses of all other rats. The average correlation for each of the other animals with the pooled response was 0.24 (± 0.09), and the correlation of this slowest animal with the others was very similar, 0.23.

To further examine the visual features that could explain rat performance, we calculated the correlation between the rat performances and image brightness of the transformations. We did this by calculating the difference in brightness of the base pair (brightness base target – brightness base distractor), and subtracting the difference in brightness of every test target–distractor pair for each test protocol (brightness test target – brightness test distractor for each test pair). We then correlated these 287 brightness values (one for each test image pair) with the average rat performance for each test image pair. We performed a similar correlation analysis for pixel similarity to investigate the correlation between pixel similarity of the test stimuli in relation to the base stimuli with the average performance of the animals on all nine test protocols. We did this by calculating the pixel similarity between the base target with every other testing distractor (A), the pixel similarity between the base target with every other testing target (B), the pixel similarity between the base distractor with every other testing distractor (C), and the pixel similarity between the base distractor with every other testing target (D). For each test image pair, we then calculated the average of (A) and (D), and subtracted the average of (C) and (B) from it. We correlated these 287 values (one for each image pair) with the average rat performance on all test image pairs.

## Computational modelling

One important goal of this study was to create a CNN-informed stimulus set to present to the animals. To do so, we followed the steps of *Schnell et al., 2023* and *Vinken and Op de Beeck, 2021* to train a CNN on the same stimuli on which our animals were trained. The steps of training the network are identical to *Schnell et al., 2023*, and a short description will follow here. We used the standard AlexNet CNN architecture that was pre-trained on ImageNet to classify images into 1000 object categories (MATLAB 2021b Deep Learning Toolbox). Following *Vinken and Op de Beeck, 2021*, we applied principal component analysis to calculate the activations in every layer to standardize the values across inputs and reduce the dimensionality. We then trained a linear support vector machine classifier by using the MATLAB function fitclinear, with limited-memory BFGS solver and default regularization. We performed this with the standardized DNN layer activations in the principal component space as inputs, before ReLU, to our 24 training stimuli (see *Figure 1*), that is, all stimuli of the *training*, *dimension* learning, and *transformations* protocols. The layers of AlexNet were divided into 13 sublayers, similar to that in *Schnell et al., 2023* and *Vinken and Op de Beeck, 2021*.

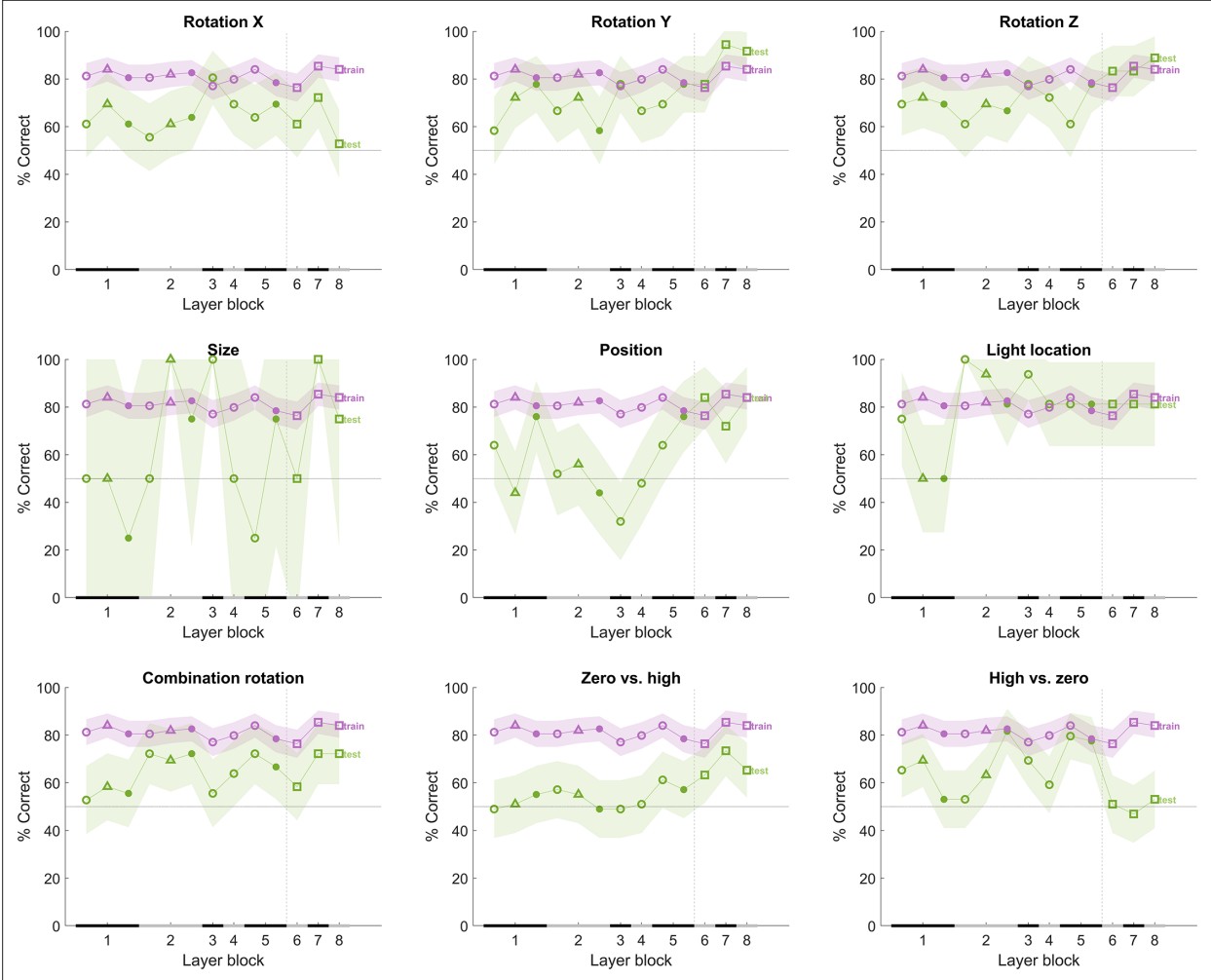

**Figure 8.** The performance of the convolutional deep neural network (CNN) after training on our training stimuli, with noise added to its input. The naming convention on the x-axis corresponds to the layers of the network, identical as in **Figure 4**. The performance (y-axis) illustrates that each layer is challenged by at least part of the test protocols. The purple line indicates the training performance, and the green line indicates the test performance of the neural network. The x-axis on each subplot indicates the block of the layer: layer blocks 1–8 correspond to (convolutional layer 1, normalization layer 1, pool layer 1), (convolutional layer 2, normalization layer 2, pool layer 2), convolutional layer 3, convolutional layer 4 (convolutional layer 5, pool layer 5), fully connected layer 6, fully connected layer 7, and fully connected layer 8, respectively. The black and grey horizontal lines on the x-axis indicate the layer blocks (block 1 consisting of conv1, norm1, pool1; block 2 consisting of conv2, norm2, pool2; block 3–4 corresponding to conv3-4, respectively; block 5 consisting of conv5, pool5; block 6-7-8 corresponding to fc6-7-8, respectively). The vertical grey dashed line indicates the division between convolutional and fully connected layer blocks. The horizontal dashed line indicates chance level. The shaded error bounds correspond to 95% confidence intervals calculated using Jackknife standard error estimates, as done previously in **Vinken and Op de Beeck, 2021**. The different markers indicate different sorts of layers: circle for convolutional layers, triangle for normalization layers, point for pool layers, and squares for fully connected layers.

*Figure 8* shows the performance of the network for each of the test protocols after training classifiers on the training stimuli using the different CNN layers. We added noise to the inputs of the network such that the average training performance, averaged over 100 iterations, lies around 75%. By adding noise in this way, the performance on the training pairs matches overall with rat performance on those pairs; otherwise, the performance of the network would be at 100% on the training pairs, and this would complicate comparisons with the animal data (see also *Vinken and Op de Beeck, 2021*). Note that the results for the *size* test are unreliable given the low number of stimulus pairs in that test. The performance of the network on the tests (green line in *Figure 8*) differs among the tests and across layers, but typically the network had no problems to achieve a training performance of about 85% in all test protocols in at least some layers. The change in performance across layers is variable across test protocols.

To examine the performance of the model for specific image pairs during training and testing in more detail than possible with a binary categorization decision, we calculate the distance to the classifier's hyperplane (decision boundary) of the targets and distractors. We do this by computing the difference in signed distance to the hyperplane between target and distractor (target – distractor). This is referred to as the classification score. For each stimulus pair in the test protocols, we computed this classification score, and we have such a score per layer.

We used this classification score to select image pairs for a CNN-informed stimulus set. To do so, we randomly chose one target and one distractor from a subset of the pool of all 4 × 4 stimuli, including all possible transformations on these stimuli. This resulted in a stimulus pool of 10.290 stimuli (5145 targets, 5145 distractors) to randomly choose two from, and 5145 * 5145 (26 471 025) possible resulting pairs of two stimuli. Once one random target and one random distractor was chosen, the DNN was tested in a similar manner as we did for the six test protocols. We performed a total of 10,000 iterations of randomly choosing a target and distractor pair. For each iteration, we calculated the average classification score of layers 1–3 and 11–13 as we wanted to compare those two levels of processing (earlier layers vs. higher layers). After these 10,000 iterations, we fine-tuned and filtered the results according to the profile of performance across earlier and higher layers (see *Supplementary file 1g*). This fine-tuning started by calculating the distribution and standard deviation for two profiles of interest, that is, (i) where early layers show an average classification score close to zero but higher layers show high classification scores (zero vs. high), and (ii) where early layers show high classification scores but higher layers show close to zero classification scores (high vs. zero). The performance was expressed relative to the distribution of values across all pairs, summarized by the standard deviation of the average target–distractor difference in classification scores of the early layers and the higher layers. We found a total of 48 stimulus pairs for these two criteria and ended up choosing 14 pairs, 7 of each criterion, that we used for the final part of the animal and human study (see lower two rows in *Figure 1*).

Afterward we also calculated the binary target–distractor CNN decision performance for the image pairs in the zero vs. high and high vs. zero tests, which is shown in *Figure 8* (bottom row). The image pairs in the zero vs. high protocol are more difficult than the other protocols, in particular for the first half of the CNN layers. In contrast, the *high* vs. *zero* protocol is the only protocol associated with chance performance in the last three layers. These analyses confirm that the CNN-based image pair selection resulted in protocols that are very different from protocols that zoom in on intuitively chosen transformations and their combinations.

Comparing the rat performances to the classification scores of the network was done by calculating the correlation across image pairs between these model scores and the rat performances averaged across animals. We concatenated the performance of the animals on all nine test protocols, as well as the distance to hyperplane of the network on all nine test protocols. Correlating these two arrays resulted in the correlations as visualized in *Figure 4*. To test whether these correlations are significant, we performed a permutation test. We permutated these arrays 1000 times, resulting in a normal distribution of permutated data per layer. We then calculated, per layer how many of the permutated values are greater than or equal to the correlation that is presented in *Figure 4* and divided this by the number of permutations.

## Human study

### Participants

Data was collected from 50 participants (average age 33.24 ± 12.23; 34 females) who participated in return for a gift voucher of 10 euros. Out of these 50 participants, 5 were excluded because of outlying behaviour during the quality check protocols (see section 'MethStimuli and protocols'). All participants had normal or corrected-to-normal vision. The experiment was approved by the ethical commission of KU Leuven (G-2020-1902R3), and each participant digitally signed an informed consent form before the start of the experiment.

### Setup

For the human part of this study, we developed an online experiment using PsychoPy3 (v2020.1.3, Python version 3.8.10) and placed it on the online platform Pavlovia. All participants received the link

and their individual participant number by e-mail with which they could participate in the experiment on their own computer. It took 30–45 min to complete the online study.

## MethStimuli and protocols

We used the same stimuli as in the animal study. The human experiment underwent the same phases as depicted in *Figure 1*, albeit with small changes. We dropped the one-third old trials in the test protocols and included two additional *dimension learning* protocols in between the first counterbalanced tests as quality check (see *Figure 1—figure supplement 1*). *Supplementary file 1h* provides an overview of the number of trials during the human experiment for each phase. *Figure 1—figure supplement 1* shows an overview of all image pairs that were presented in the human study.

Similar to that in *Bossens and Op de Beeck, 2016*, we presented the targets and distractors briefly to the left and right sides of a white fixation cross on a grey background. Each stimulus was presented for three frames, followed by a mask (a noise image with 1/f frequency spectrum for three frames). We used this fast and eccentric stimulus presentation with a mask to resemble the stimulus perception more closely to that of rats. *Vermaercke and Op de Beeck, 2012* have found that human visual acuity in these fast and eccentric presentations is not significantly better than the reported visual acuity of rats. By using this approach, we avoid that differences in strategies between humans and rats would be explained by such a difference in acuity. Participants could then answer using the 'f' and 'j' keys to indicate which position they thought was the correct position. If they thought the target was on the left side of the fixation cross, they had to press 'f', and press 'j' if they thought the target was on the right side. Participants received feedback during the shaping and the three training phases. This happened by colouring the fixation cross green if they answered correctly and red if they answered incorrectly. Each trial was followed by an ITI of 0.5 s. During the *shaping* and *training* phase, we kept a running average of the past 20 (*shaping*) and 40 (*training*) trials and participants continued to the next phase when they reached a performance of 80% or higher on the last 20 or 40 trials, similar to that in *Bossens and Op de Beeck, 2016*. There was no time limit for the participants for providing a response. The order of the first six test protocols (*rotation X, rotation Y, rotation Z, size, light location,* and *position*) was counterbalanced between the participants based on the participant number, as well as the order of the last two test protocols (*zero* vs. *high* and *high* vs. *zero*), similar to the approach in the rat study. *Supplementary file 1f* indicates the average number of trials per test protocol for all human participants together.

In terms of instructions, we explained to participants that they would see two figures appearing at the same time very quickly next to a fixation cross, and they would have to make a decision of which figure is the correct one. We mentioned that during training the fixation cross would turn green if they answered correctly and red if they answered incorrectly. Participants were informed that during testing they would not get feedback (changing colour of the fixation cross) anymore and that they would have to use the knowledge they gained throughout training to make their decision in the testing.

We performed a similar correlation analysis as with rat performance to investigate the correlation between pixel similarity and brightness with human performance. We followed the exact same steps as we did for rat performance.

# Additional information

## Funding

| Funder | Grant reference number | Author |
| --- | --- | --- |
| Excellence of Science | G0E8718N | Hans Op de Beeck |
| KU Leuven infrastructure grants | AKUL/13/06 | Hans Op de Beeck |
| KU Leuven infrastructure grants | AKUL/19/05 | Hans Op de Beeck |
| KU Leuven Research Council Project | C14/16/031 | Hans Op de Beeck |

| Funder | Grant reference number | Author |
| --- | --- | --- |
| KU Leuven Research Council Project | C14/21/047 | Hans Op de Beeck |

The funders had no role in study design, data collection and interpretation, or the decision to submit the work for publication.

## Author contributions

Anna Elisabeth Schnell, Conceptualization, Data curation, Formal analysis, Investigation, Visualization, Methodology, Writing - original draft, Writing - review and editing; Maarten Leemans, Kasper Vinken, Writing - review and editing; Hans Op de Beeck, Conceptualization, Data curation, Supervision, Funding acquisition, Investigation, Methodology, Project administration, Writing - review and editing

## Author ORCIDs

Anna Elisabeth Schnell ⓘ http://orcid.org/0000-0001-9278-1817
Kasper Vinken ⓘ http://orcid.org/0000-0001-7038-9638

## Ethics

Human subjects: The experiment was approved by the ethical commission of KU Leuven (G-2020-1902-R3) and each participant digitally signed an informed consent before the start of the experiment.This study followed the latest update of the Declaration of Helsinki (World Medical Association, 2013).

All experiments and procedures involving living animals were approved by the Ethical Committee for animal experimentation of the KU Leuven and were in accordance with the European Commission Directive of September 22, 2010 (2010/63/EU). We have reported the study in accordance with the ARRIVE guidelines.

Reviewer #1 (Public Review): https://doi.org/10.7554/eLife.87719.3.sa1
Reviewer #2 (Public Review): https://doi.org/10.7554/eLife.87719.3.sa2
Author response https://doi.org/10.7554/eLife.87719.3.sa3

# Additional files

## Supplementary files

• Supplementary file 1. Supplementary tables. (**a**) Results of binomial test on the six test protocols with the pooled data of all animals together, on the old trials and the new trials. (**b**) Marginal means and standard deviation of the rotation X and rotation Z protocols. (**c**) Correlation between neighbouring layers of the deep neural network. (**d**) Results of the linear regression model with rat performances. (**e**) Results of the linear regression model with human performances. (**f**) Average number of trials (SD) per test protocol and per stimulus pair (SP). (**g**) The two criteria of choosing a CNN-informed stimulus set. (**h**) Overview of the human experiment. (**i**) An overview of the performance of the animals on the first six test protocols.

• Supplementary file 2. Information about the pilot study.

• MDAR checklist

## Data availability

The data has been made publicly available via the Open Science Framework and can be accessed at https://osf.io/9eqyz/.

The following datasets were generated:

| Author(s) | Year | Dataset title | Dataset URL | Database and Identifier |
|---|---|---|---|---|
| Schnell AE, Op de Beeck H, Vinken K, Leemans M | 2023 | Multidimensional stimulus set | https://doi.org/10.17605/OSF.IO/V9WBN | Open Science Framework, 10.17605/OSF.IO/V9WBN |
| Schnell AE, Op de Beeck H, Vinken K, Leemans M | 2023 | A computationally informed comparison between the strategies of rodents and humans in visual object recognition | https://doi.org/10.17605/OSF.IO/9EQYZ | Open Science Framework, 10.17605/OSF.IO/9EQYZ |

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
