## [Editor Report · eLife assessment]

Schnell et al. report **important** differences between the strategies used by rodents and humans when discriminating different visual objects. The evidence supporting these findings is **convincing**, showing that rat performance was influenced far more by low-level cues compared to humans. It is, however, unclear to what extent these differences can be explained by the lower visual acuity of rats. This work will be of general interest to vision and cognition researchers, particularly those studying object vision.

---

## [Referee Report · Reviewer #1 (Public Review)]

Schnell et al. performed two extensive behavioral experiments concerning the processing of objects in rats and humans. To this aim, they designed a set of objects parametrically varying along alignment and concavity and then they used activations from a pretrained deep convolutional neural network to select stimuli that would require one of two different discrimination strategies, i.e. relying on either low- or high-level processing exclusively. The results show that rodents rely more on low-level processing than humans.

Strengths:

1. The results are challenging and call for a different interpretation of previous evidence. Indeed, this work shows that common assumptions about task complexity and visual processing are probably biased by our personal intuitions and are not equivalent in rodents, which instead tend to rely more on low-level properties.

2. This is an innovative (and assumption-free) approach that will prove useful to many visual neuroscientists. Personally, I second the authors' excitement about the proposed approach, and its potential to overcome the limits of experimenters' creativity and intuitions. In general, the claims seem well supported and the effects sufficiently clear.

3. This work provides an insightful link between rodent and human literature on object processing. Given the increasing number of studies on visual perception involving rodents, these kinds of comparisons are becoming crucial.

4. The paper raises several novel questions that will prompt more research in this direction.

Weaknesses:

1. The choice of alignment and concavity as baseline properties of the stimuli is not properly discussed.

2. From the low-correlations I got the feeling that AlexNet is not the best baseline model for rat visual processing.

---

## [Referee Report · Reviewer #2 (Public Review)]

Schnell and colleagues trained rats on a two-alternative forced choice visual discrimination task. They used object pairs that differed in their concavity and the alignment of features. They found that rats could discriminate objects across various image transformations. Rat performance correlated best with late convolutional layers of an artificial neural network and was partially explained by factors of brightness and pixel-level similarity. In contrast, human performance showed the strongest correlation with higher, fully connected layers, indicating that rats employed simpler strategies to accomplish this task as compared to humans.

Strengths:

1. This is a methodologically rigorous study. The authors tested a substantial number of rats across a large variety of stimuli.

2. The innovative use of neural networks to generate stimuli with varying levels of complexity is a compelling approach that motivates principled experimental design.

3. The study provides important data points for cross-species comparisons of object discrimination behavior

4. The data strongly support the authors' conclusion that rats and humans rely on different visual features for discrimination tasks.

5. This is a valuable study that provides novel, important insights into the visual capabilities of rats.

Weaknesses:

1. The impact of rat visual acuity (~1cycle/degree) on the discriminability of stimuli could be more directly modeled and taken into consideration when comparing rat behavior to humans, who possess substantially higher acuity.

2. The distinction between low- and high-level visual behavior is coarse, and it remains uncertain which specific features rats utilized for discrimination. The correlations with brightness and pixel-level similarity do provide some insight.

3. The relatively weak correspondence between rat behavior and AlexNet raises the question of which network architecture, whether computational or biological, might better capture rat behavior, particularly to the level of cross-rat consistency.

---

## [Author Response]

The following is the authors’ response to the current reviews.

We thank the editor for the eLife assessment and reviewers for their remaining comments. We will address them in this response.

First, we thank eLife for the positive assessment. Regarding the point of visual acuity that is mentioned in this assessment, we understand that this comment is made. It is not an uncommon comment when rodent vision is discussed. However, we emphasize that we took the lower visual acuity of rats and the higher visual acuity of humans into account when designing the human study, by using a fast and eccentric stimulus presentation for humans. As a result, we do not expect a higher discriminability of stimuli in humans. We have described this in detail in our Methods section when describing the procedure in the human experiment:

“We used this fast and eccentric stimulus presentation with a mask to resemble the stimulus perception more closely to that of rats. Vermaercke & Op de Beeck (2012) have found that human visual acuity in these fast and eccentric presentations is not significantly better than the reported visual acuity of rats. By using this approach we avoid that differences in strategies between humans and rats would be explained by such a difference in acuity”

Second, regarding the remaining comment of Reviewer #2 about our use of AlexNet:

While it is indeed relevant to further look into different computational architectures, we chose to not do this within the current study. First, it is a central characteristic of the study procedure that the computational approach and chosen network is chosen early on as it is used to generate the experimental design that animals are tested with. We cannot decide after data collection to use a different network to select the stimuli with which these data were collected. Second, as mentioned in our first response, using AlexNet is not a random choice. It has been used in many previously published vision studies that were relatively positive about the correspondence with biological vision (Cadieu et al., 2014; Groen et al., 2018; Kalfas et al., 2018; Nayebi et al., 2023; Zeman et al., 2020). Third, our aim was not to find a best DNN model for rat vision, but instead examining the visual features that play a role in our complex discrimination task with a model that was hopefully a good enough starting point. The fact that the designs based upon AlexNet resulted in differential and interpretable effects in rats as well as in humans suggests that this computational model was a good start. Comparing the outcomes of different networks would be an interesting next step, and we expect that our approach could work even better when using a network that is more specifically tailored to mimic rat visual processing.

Finally, regarding the choice to specifically chose alignment and concavity as baseline properties, this choice is probably not crucial for the current study. We have no reason to expect rats to have an explicit notion about how a shape is built up in terms of a part-based structure, where alignment relates to the relative position of the parts and concavity is a property of the main base. For human vision it might be different, but we did not focus on such questions in this study.

The following is the authors’ response to the original reviews.

We would like to thank you for giving us the opportunity to submit a revised draft our manuscript. We appreciate the time and effort that you dedicated to providing insightful feedback on our manuscript and are grateful for the valuable comments and improvements on our paper. It helped us to improve our manuscript. We have carefully considered the comments and tried our best to address every one of them. We have added clarifications in the Discussion concerning the type of neural network that we used, about which visual features might play a role in our results as well as clarified the experimental setup and protocol in the Methods section as these two sections were lacking key information points.

Below we provide a response to the public comments and concerns of the reviewers.

Several key points were addressed by at least two reviewers, and we will respond to them first.

A first point concerns the type of network we used. In our study, we used AlexNet to simulate the ventral visual stream and to further examine rat and human performance. While other, more complex neural networks might lead to other results, we chose to work with AlexNet because it has been used in many other vision studies that are published in high impact journals (Cadieu et al., 2014; Groen et al., 2018; Kalfas et al., 2018; Nayebi et al., 2023; Zeman et al., 2020). We did not try to find a best DNN model for rat vision but instead, we were looking for an explanation of which visual features play a role in our complex discrimination task. We added a consideration to our Discussion addressing why we worked with AlexNet. Since our data will be published on OSF, we encourage to researchers to use our data with other, more complex neural networks and to further investigate this issue.

A second point that was addressed by multiple reviewers concerns the visual acuity of the animals and its impact on their performance. The position of the rat was not monitored in the setup. In a previous study in our lab (Crijns & Op de Beeck, 2019), we investigated the visual acuity of rats in the touchscreen setups by presenting gratings with different cycles per screen to see how it affects their performance in orientation discrimination. With the results from this study and general knowledge about rat visual acuity, we derived that the decision distance of rats lies around 12.5cm from the screen. We have added this paragraph to the Discussion.

A third key point that needs to be addressed as a general point involves which visual features could explain rat and human performance. We reported marked differences between rat and human data in how performance varied across image trials, and we concluded through our computationally informed tests and analyses that rat performance was explained better by lower levels of processing. Yet, we did not investigate which exact features might underlie rat performance. As a starter, we have focused on taking a closer look at pixel similarity and brightness and calculating the correlation between rat/human performance and these two visual features.

We calculated the correlation between the rat performances and image brightness of the transformations. We did this by calculating the difference in brightness of the base pair (brightness base target – brightness base distractor), and subtracting the difference in brightness of every test target-distractor pair for each test protocol (brightness test target – brightness test distractor for each test pair). We then correlated these 287 brightness values (1 for each test image pair) with the average rat performance for each test image pair. This resulted in a correlation of 0.39, suggesting that there is an influence of brightness in the test protocols. If we perform the same correlation with the human performances, we get a correlation of -0.12, suggesting a negative influence of brightness in the human study.

We calculated the correlation between pixel similarity of the test stimuli in relation to the base stimuli with the average performance of the animals on all nine test protocols. We did this by calculating the pixel similarity between the base target with every other testing distractor (A), the pixel similarity between the base target with every other testing target (B), the pixel similarity between the base distractor with every other testing distractor (C) and the pixel similarity between the base distractor with every other testing target (D). For each test image pair, we then calculated the average of (A) and (D), and subtracted the average of (C) and (B) from it. We correlated these 287 values (one for each image pair) with the average rat performance on all test image pairs, which resulted in a correlation of 0.34, suggesting an influence of pixel similarity in rat behaviour. Performing the same correlation analysis with the human performances results in a correlation of 0.12.

We have also addressed this in the Discussion of the revised manuscript. Note that the reliability of the rat data was 0.58, clearly higher than the correlations with brightness and pixel similarity, thus these features capture only part of the strategies used by rats.

We have also responded to all other insightful suggestions and comments of the reviewers, and a point-by-point response to the more major comments will follow now.

**Reviewer #1, general comments:**
The authors should also discuss the potential reason for the human-rat differences too, and importantly discuss whether these differences are coming from the rather unusual approach of training used in rats (i.e. to identify one item among a single pair of images), or perhaps due to the visual differences in the stimuli used (what were the image sizes used in rats and humans?). Can they address whether rats trained on more generic visual tasks (e.g. same-different, or category matching tasks) would show similar performance as humans?

The task that we used is typically referred to as a two-alternative forced choice (2AFC). This is a simple task to learn. A same-different task is cognitively much more demanding, also for artificial neural networks (see e.g. Puebla & Bowers, 2022, J. Vision). A one-stimulus choice task (probably what the reviewer refers to with category matching) is known to be more difficult compared to 2AFC, with a sensitivity that is predicted to be Sqrt(2) lower according to signal detection theory (MacMillan & Creelman, 1991). We confirmed this prediction empirically in our lab (unpublished observations). Thus, we predict that rats perform less good in the suggested alternatives, potentially even (in case of same-different) resulting in a wider performance gap with humans.

I also found that a lot of essential information is not conveyed clearly in the manuscript. Perhaps it is there in earlier studies but it is very tedious for a reader to go back to some other studies to understand this one. For instance, the exact number of image pairs used for training and testing for rats and humans was either missing or hard to find out. The task used on rats was also extremely difficult to understand. An image of the experimental setup or a timeline graphic showing the entire trial with screenshots would have helped greatly.

All the image pairs used for training and testing for rats and humans are depicted in Figure 1 (for rats) and Supplemental Figure 6 (for humans). For the first training protocol (Training), only one image pair was shown, with the target being the concave object with horizontal alignment of the spheres. For the second training protocol (Dimension learning), three image pairs were shown, consisting of the base pair, a pair which differs only in concavity, and a pair which differs only in alignment. For the third training protocol (Transformations) and all testing protocols, all combination of targets and distractors were presented. For example, in the Rotation X protocol, the stimuli consisted of 6 targets and 6 distractors, resulting in a total of 36 image pairs for this protocol.The task used on rats is exactly as shown in Figure 1. A trial started with two blank screens. Once the animal initiated a trial by sticking its head in the reward tray, one stimulus was presented on each screen. There was no time limit and so the stimuli remained on the screen until the animal made a decision. If the animal touched the target, it received a sugar pellet as reward and a ITI of 20s started. If the animal touched the distractor, it did not receive a sugar pellet and a time-out of 5s started in addition to the 20s ITI.

We have clarified this in the manuscript.

The authors state that the rats received random reward on 80% of the trials, but is that on 80% of the correctly responded trials or on 80% of trials regardless of the correctness of the response? If these are free choice experiments, then the task demands are quite different. This needs to be clarified. Similarly, the authors mention that 1/3 of the trials in a given test block contained the old base pair - are these included in the accuracy calculations?

The animals receive random reward on 80% on all testing trials with new stimuli, regardless of the correctness of the response. This was done to ensure that we can measure true generalization based upon learning in the training phase, and that the animals do not learn/are not trained in these testing stimuli. For the trials with the old stimuli (base pair), the animals always received real reward (reward when correct; no reward in case of error).

The 1/3rd trials with old stimuli are not included in the accuracy calculations but were used as a quality check/control to investigate which sessions have to be excluded and to assure that the rats were still doing the task properly. We have added this in the manuscript.

The authors were injecting noise with stimuli to cDNN to match its accuracy to rat. However, that noise potentially can interacted with the signal in cDNN and further influence the results. That could generate hidden confound in the results. Can they acknowledge/discuss this possibility?

Yes, adding noise can potentially interact with the signal and further influence the results. Without noise, the average training data of the network would lie around 100% which would be unrealistic, given the performances of the animals. To match the training performance of the neural networks with that of the rats, we added noise 100 times and averaged over these iterations (cfr. (Schnell et al., 2023; Vinken & Op de Beeck, 2021)).

**Reviewer #2, weaknesses:**
1. There are a few inconsistencies in the number of subjects reported. Sometimes 45 humans are mentioned and sometimes 50. Probably they are just typos, but it's unclear.

Thank you for your feedback. We have doublechecked this and changed the number of subjects where necessary. We collected data from 50 human participants, but had to exclude 5 of them due to low performance during the quality check (Dimension learning) protocols. Similarly, we collected data from 12 rats but had to exclude one animal because of health issues. All these data exclusion steps were mentioned in the Methods section of the original version of the manuscript, but the subject numbers were not always properly adjusted in the description in the Results section. This is now corrected.

1. A few aspects mentioned in the introduction and results are only defined in the Methods thus making the manuscript a bit hard to follow (e.g. the alignment dimension), thus I had to jump often from the main text to the methods to get a sense of their meaning.

Thank you for your feedback. We have clarified some aspects in the Introduction, such as the alignment dimension.

1. Many important aspects of the task are not fully described in the Methods (e.g. size of the stimuli, reaction times and basic statistics on the responses).

We have added the size of the stimuli to the Methods section and clarified that the stimuli remained on the screen until the animals made a choice. Reaction time in our task would not be interpretable given that stimuli come on the screen when the animal initiates a trial with its back to the screen. Therefore we do not have this kind of information.

**Reviewer #1**
• Can the authors show all the high vs zero and zero vs high stimulus pairs either in the main or supplementary figures? It would be instructive to know if some other simple property covaried between these two sets.

In Figure 1, all images of all protocols are shown. For the High vs. Zero and Zero vs. High protocols, we used a deep neural network to select a total of 7 targets and 7 distractors. This results in 49 image pairs (every combination of target-distractor).

• Are there individual differences across animals? It would be useful for the authors to show individual accuracy for each animal where possible.

We now added individual rat data for all test protocols – 1 colour per rat, black circle = average. We have added this picture to the Supplementary material (Supplementary Figure 1).

• Figure 1 - it was not truly clear to me how many image pairs were used in the actual experiment. Also, it was very confusing to me what was the target for the test trials. Additionally, authors reported their task as a categorisation task, but it is a discrimination task.

Figure 1 shows all the images that were used in this study. Every combination of every target-distractor in each protocol (except for Dimension learning) was presented to the animals. For example in Rotation X, the test stimuli as shown in Fig. 1 consisted of 6 targets and 6 distractors, resulting in a total of 36 image pairs for this test protocol.

In each test protocol, the target corresponded to the concave object with horizontally attached spheres, or the object from the pair that in the stimulus space was closed to this object. We have added this clarification in the Introduction: “We started by training the animals in a base stimulus pair, with the target being the concave object with horizontally aligned spheres. Once the animals were trained in this base stimulus pair, we used the identity-preserving transformations to test for generalization.” as well as in the caption of Figure 1. We have changed the term “categorisation task” to “discrimination task” throughout the manuscript.

• Figure 2 - what are the red and black lines? How many new pairs are being tested here? Panel labels are missing (a/b/c etc)

We have changed this figure by adding panel labels, and clarifying the missing information in the caption. All images that were shown to the animals are presented on this figure. For Dimension Learning, only three image pairs were shown (base pair, concavity pair, alignment pair) and for the Transformations protocol, every combination of every target and distractor were shown, i.e. 25 image pairs in total.

• Figure 3 - last panel: the 1st and 2nd distractor look identical.

We understand your concern as these two distractors indeed look quite similar. They are different however in terms of how they are rotated along the x, y and z axes (see Author response image 1 for a bigger image of these two distractors). The similarity is due to the existence of near-symmetry in the object shape which causes high self-similarity for some large rotations.

• Line 542 – authors say they have ‘concatenated’ the performance of the animals, but do they mean they are taking the average across animals?

It is both. In this specific analysis we calculated the performance of the animals, which was indeed averaged across animals, per test protocol, per stimulus pair. This resulted in 9 arrays (one for each test protocol) of several performances (1 for each stimulus pair). These 9 arrays were concatenated by linking them together in one big array (i.e. placing them one after the other). We did the same concatenation with the distance to hyperplane of the network on all nine test protocols. These two concatenated arrays with 287 values each (one with the animal performance and one with the DNN performance) were correlated.

• Line 164 - What are these 287 image pairs - this is not clear.

The 287 image pairs correspond to all image pairs of all 9 test protocols: 36 (Rotation X) + 36 (Rotation Y) + 36 (Rotation Z) + 4 (Size) + 25 (Position) + 16 (Light location) + 36 (Combination Rotation) + 49 (Zero vs. high) + 49 (High vs. zero) = 287 image pairs in total. We have clarified this in the manuscript.

• Line 215 - Human rat correlation (0.18) was comparable to the best cDNN layer correlation. What does this mean?

The human rat correlation (0.18) was closest to the best cDNN layer - rat correlation (about 0.15). In the manuscript we emphasize that rat performance is not well captured by individual cDNN layers.

**Reviewer #2**
Major comments• In l.23 (and in the methods) the authors mention 50 humans, but in l.87 they are 45. Also, both in l.95 and in the Methods the authors mention "twelve animals" but they wrote 11 elsewhere (e.g. abstract and first paragraph of the results).

In our human study design, we introduced several Dimension learning protocols. These were later used as a quality check to indicate which participants were outliers, using outlier detection in R. This resulted in 5 outlying human participants, and thus we ended with a pool of 45 human participants that were included in the analyses. This information was given in the Methods section of the original manuscript, but we did not mention the correct numbers everywhere. We have corrected this in the manuscript. We also changed the number of participants (humans and rats) to the correct one throughout the entire manuscript.

• At l.95 when I first met the "4x4 stimulus grid" I had to guess its meaning. It would be really useful to see the stimulus grid as a panel in Figure 1 (in general Figures S1 and S4 could be integrated as panels of Figure 1). Also, even if the description of the stimulus generation in the Methods is probably clear enough, the authors might want to consider adding a simple schematic in Figure 1 as well (e.g. show the base, either concave or convex, and then how the 3 spheres are added to control alignment).

We have added the 4x4 stimulus grid in the main text.

• There is also another important point related to the choice of the network. As I wrote, I find the overall approach very interesting and powerful, but I'm actually worried that AlexNet might not be a good choice. I have experience trying to model neuronal responses from IT in monkeys, and there even the higher layers of AlexNet aren't that helpful. I need to use much deeper networks (e.g. ResNet or GoogleNet) to get decent fits. So I'm afraid that what is deemed as "high" in AlexNet might not be as high as the authors think. It would be helpful, as a sanity check, to see if the authors get the same sort of stimulus categories when using a different, deeper network.

We added a consideration to the manuscript about which network to use (see the Discussion): “We chose to work with Alexnet, as this is a network that has been used as a benchmark in many previous studies (e.g. (Cadieu et al., 2014; Groen et al., 2018; Kalfas et al., 2018; Nayebi et al., 2023; Zeman et al., 2020)), including studies that used more complex stimuli than the stimulus space in our current study. […] . It is in line with the literature that a typical deep neural network, AlexNet and also more complex ones, can explain human and animal behaviour to a certain extent but not fully. The explained variance might differ among DNNs, and there might be DNNs that can explain a higher proportion of rat or human behaviour. Most relevant for our current study is that DNNs tend to agree in terms of how representations change from lower to higher hierarchical layers, because this is the transformation that we have targeted in the Zero vs. high and High vs. zero testing protocols. (Pinto et al., 2008) already revealed that a simple V1-like model can sometimes result in surprisingly good object recognition performance. This aspect of our findings is also in line with the observation of Vinken & Op de Beeck (2021) that the performance of rats in many previous tasks might not be indicative of highly complex representations. Nevertheless, there is still a relative difference in complexity between lower and higher levels in the hierarchy. That is what we capitalize upon with the Zero vs. high and High vs. zero protocols. Thus, it might be more fruitful to explicitly contrast different levels of processing in a relative way rather than trying to pinpoint behaviour to specific levels of processing.”

• The task description needs way more detail. For how long were the stimuli presented? What was their size? Were the positions of the stimuli randomized? Was it a reaction time task? Was the time-out used as a negative feedback? In case, when (e.g. mistakes or slow responses)? Also, it is important to report some statistics about the basic responses. What was the average response time, what was the performance of individual animals (over days)? Did they show any bias for a particular dimension (either the 2 baseline dimensions or the identity preserving ones) or side of response? Was there a correlation within animals between performance on the baseline task and performance on the more complex tasks?

Thank you for your feedback. We have added more details to the task description in the manuscript.

The stimuli were presented on the screens until the animals reacted to one of the two screens. The size of the stimuli was 100 x 100 pixel. The position of the stimuli was always centred/full screen on the touchscreens. It was not a reaction time task and we also did not measure reaction time.

• Related to my previous comment, I wonder if the relative size/position of the stimulus with respect to the position of the animal in the setup might have had an impact on the performance, also given the impact of size shown in Figure 2. Was the position of the rat in the setup monitored (e.g. with DeepLabCut)? I guess that on average any effect of the animal position might be averaged away, but was this actually checked and/or controlled for?

The position of the rat was not monitored in the setup. In a previous study from our lab (Crijns & Op de Beeck, 2019), we investigated the visual acuity of rats in the touchscreen setups by presenting gratings with different cycles per screen to see how it affects their performance in orientation discrimination. With the results from this study and general knowledge about rat visual acuity, we derived that the decision distance of rats lies around 12.5cm from the screen. We have added this to the discussion.

Minor comments• l.33 The sentence mentions humans, but the references are about monkeys. I believe that this concept is universal enough not to require any citation to support it.

Thank you for your feedback. We have removed the citations.

• This is very minor and totally negligible. The acronymous cDNN is not that common for convents (and it's kind of similar to cuDNN), it might help clarity to stick to a more popular acronymous, e.g. CNN or ANN. Also, given that the "high" layers used for stimulus selection where not convolutional layers after all (if I'm not mistaken).

Thank you for your feedback. We have changed the acronym to ‘CNN’ in the entire manuscript.

• In l.107-109 the authors identified a few potential biases in their stimuli, and they claim these biases cannot explain the results. However, the explanation is given only in the next pages. It might help to mention that before or to move that paragraph later, as I was just wondering about it until I finally got to the part on the brightness bias.

We expanded the analysis of these dimensions (e.g. brightness) throughout the manuscript.

• It would help a lot the readability to put also a label close to each dimension in Figures 2 and 3. I had to go and look at Figure S4 to figure that out.

Figures 2 and 3 have been updated, also including changes related to other comments.

• In Figure 2A, please specify what the red dashed line means.

We have edited the caption of Figure 2: “Figure 2 (a) Results of the Dimension learning training protocol. The black dashed horizontal line indicates chance level performance and the red dashed line represents the 80% performance threshold. The blue circles on top of each bar represent individual rat performances. The three bars represent the average performance of all animals on the old pair (Old), the pair that differs only in concavity (Conc) and on the pair that differs only in alignment (Align). (b) Results of the Transformations training protocol. Each cell of the matrix indicates the average performance per stimulus pair, pooled over all animals. The columns represent the distractors, whereas the rows separate the targets. The colour bar indicates the performance correct. ”

• Related to that, why performing a binomial test on 80%? It sounds arbitrary.

We performed the binomial test on 80% as 80% is our performance threshold for the animals

• The way the cDNN methods are introduced makes it sound like the authors actually fine-tuned the weights of AlexNet, while (if I'm not mistaken), they trained a classifier on the activations of a pre-trained AlexNet with frozen weights. It might be a bit confusing to readers. The rest of the paragraph instead is very clear and easy to follow.

We think the most confusing sentence was “ Figure 7 shows the performance of the network after training the network on our training stimuli for all test protocols. “ We changed this sentence to “ Figure 8 shows the performance of the network for each of the test protocols after training classifiers on the training stimuli using the different DNN layers.“

**Reviewer #3**
Main recommendations:Although it may not fully explain the entire pattern of visual behavior, it is important to discuss rat visual acuity and its impact on the perception of visual features in the stimulus set.

We have added a paragraph to the Discussion that discusses the visual acuity of rats and its impact on perceiving the visual features of the stimuli.

The authors observed a potential influence of image brightness on behavior during the dimension learning protocol. Was there a correlation between image brightness and the subsequent image transformations?

We have added this to the Discussion: “To further investigate to which visual features the rat performance and human performance correlates best with, we calculated the correlation between rat performance and pixel similarity of the test image pairs, as well as the correlation between rat performance and brightness in the test image pairs. Here we found a correlation of 0.34 for pixel similarity and 0.39 for brightness, suggesting that these two visual features partly explain our results when compared to the full-set reliability of rat performance (0.58). If we perform the same correlation with the human performances, we get a correlation of 0.12 for pixel similarity and -0.12 for brightness. With the full-set reliability of 0.58 (rats) and 0.63 (humans) in mind, this suggests that even pixel similarity and brightness only partly explain the performances of rats and humans.”

Did the rats rely on consistent visual features to perform the tasks? I assume the split-half analysis was on data pooled across rats. What was the average correlation between rats? Were rats more internally consistent (split-half within rat) than consistent with other rats?

The split-half analysis was indeed performed on data pooled across rats. We checked whether rats are more internally consistent by comparing the split-half within correlations with the split-half between correlations. For the split-half within correlations, we split the data for each rat in two subsets and calculated the performance vectors (performance across all image pairs). We then calculated the correlation between these two vectors for each animal. To get the split-half between correlation, we calculated the correlation between the performance vector of every subset data of every rat with every other subset data from the other rats. Finally, we compared for each animal its split-half within correlation with the split-half between correlations involving that animal. The result of this paired t-test (p = 0.93, 95%CI [-0.09; 0.08]) suggests that rats were not internally more consistent.

Discussion of the cDNN performance and its relation to rat behavior could be expanded and clarified in several ways:• The paper would benefit from further discussion regarding the low correlations between rat behavior and cDNN layers. Is the main message that cDNNs are not a suitable model for rat vision? Or can we conclude that the peak in mid layers indicates that rat behavior reflects mid-level visual processing? It would be valuable to explore what we currently know about the organization of the rat visual cortex and how applicable these models are to their visual system in terms of architecture and hierarchy.

We added a consideration to the manuscript about which network to use (see Discussion).

• The cDNN exhibited above chance performance in various early layers for several test protocols (e.g., rotations, light location, combination rotation). Does this limit the interpretation of the complexity of visual behavior required to perform these tasks?

This is not uncommon to find. Pinto et al. (2008) already revealed that a simple V1-like model can sometimes result in surprisingly good object recognition performance. This aspect of our findings is also in line with the observation of Vinken & Op de Beeck (2021) that the performance of rats in many previous tasks might not be indicative of highly complex representations. Nevertheless, there is still a relative difference in complexity between lower and higher levels in the hierarchy. That is what we capitalize upon with the High vs zero and the Zero vs high protocols. Thus, it might be more fruitful to explicitly contrast different levels of processing in a relative way rather than trying to pinpoint behavior to specific levels of processing. This argumentation is added to the Discussion section.

• How representative is the correlation profile between cDNN layers and behavior across protocols? Pooling stimuli across protocols may be necessary to obtain stable correlations due to relatively modest sample numbers. However, the authors could address how much each individual protocol influences the overall correlations in leave-one-out analyses. Are there protocols where rat behavior correlates more strongly with higher layers (e.g., when excluding zero vs. high)?

We prefer to base our conclusions mostly on the pooled analyses rather than individual protocols. As the reviewer also mentions, we can expect that the pooled analyses will provide the most stable results. For information, we included leave-one-out analyses in the supplemental material. Excluding the Zero vs. High protocol did not result in a stronger correlation with the higher layers. It was rare to see correlations with higher layers, and in the one case that we did (when excluding High versus zero) the correlations were still higher in several mid-level layers.

**Author response image 2. sa3fig2:** 

• The authors hypothesize that the cDNN results indicate that rats rely on visual features such as contrast. Can this link be established more firmly? e.g., what are the receptive fields in the layers that correlate with rat behavior sensitive to?

This hypothesis was made based on previous in-lab research (Schnell et al., 2023) where we found rats indeed rely on contrast features. In this study, we performed a face categorization task, parameterized on contrast features, and we investigated to what extent rats use contrast features to perform in a face categorization task. Similarly as in the current study, we used a DNN that as trained and tested on the same stimuli as the animals to investigate the representations of the animals. There, we found that the animals use contrast features to some extent and that this correlated best with the lower layers of the network. Hence, we would say that the lower layers correlate best with rat behaviour that is sensitive to contrast. Earlier layers of the network include local filters that simulate V1-like receptive fields. Higher layers of the network, on the other hand, are used for object selectivity.

• There seems to be a disconnect between rat behavior and the selection of stimuli for the high (zero) vs. zero (high) protocols. Specifically, rat behavior correlated best with mid layers, whereas the image selection process relied on earlier layers. What is the interpretation when rat behavior correlates with higher layers than those used to select the stimuli?

We agree that it is difficult to pinpoint a particular level of processing, and it might be better to use relative terms: lower/higher than. This is addressed in the manuscript by the edit in response to three comments back.

• To what extent can we attribute the performance below the ceiling for many protocols to sensory/perceptual limitations as opposed to other factors such as task structure, motivation, or distractibility?

We agree that these factors play a role in the overall performance difference. In Figure 5, the most right bar shows the percentage of all animals (light blue) vs all humans (dark blue) on the old pair that was presented during the testing protocol. Even here, the performance of the animals was lower than humans, and this pattern extended to the testing protocols as well. This was most likely due to motivation and/or distractibility which we know can happen in both humans and rats but affects the rat results more with our methodology.

Minor recommendations:• What was the trial-to-trial variability in the distance and position of the rat's head relative to the stimuli displayed on the screen? Can this variability be taken into account in the size and position protocols? How meaningful is the cDNN modelling of these protocols considering that the training and testing of the model does not incorporate this trial-to-trial variability?

We have no information on this trial-to-trial variability. We have information though on what rats typically do overall from an earlier paper that was mentioned in response to an earlier comment (Crijns et al.).

We have added a disclaimer in the Discussion on our lack of information on trial-to-trial variability.

• Several of the protocols varied a visual feature dimension (e.g., concavity & alignment) relative to the base pair. Did rat performance correlate with these manipulations? How did rat behavior relate to pixel dissimilarity, either between target and distractor or in relation to the trained base pair?

We have added this to the Discussion. See also our general comments in the Public responses.

• What could be the underlying factor(s) contributing to the difference in accuracy between the "small transformations" depicted in Figure 2 and some of the transformations displayed in Figure 3? In particular, it seems that the variability of targets and distractors is greater for the "small transformations" in Figure 2 compared to the rotation along the y-axis shown in Figure 3.

There are several differences between these protocols. Before considering the stimulus properties, we should take into account other factors. The Transformations protocol was a training protocol, meaning that the animals underwent several sessions in this protocol, always receiving real reward during the trials, and only stopping once a high enough performance was reached. For the protocols in Figure 3, the animals were also placed in these protocols for multiple sessions in order to obtain enough trials, however, the difference here is that they did not receive real reward and testing was also stopped if performance was still low.

• In Figure 3, it is unclear which pairwise transformation accuracies were above chance. It would be helpful if the authors could indicate significant cells with an asterisk. The scale for percentage correct is cut off at 50%. Were there any instances where the behaviors were below 50%? Specifically, did the rats consistently choose the wrong option for any of the pairs?It would be helpful to add "old pair", "concavity" and "alignment" to x-axis labels in Fig 2A .

We have added “old”, “conc” and “align” to the x-axis labels in Figure 2A.

• Considering the overall performance across protocols, it seems overstated to claim that the rats were able to "master the task."

When talking about “mastering the task”, we talk about the training protocols where we aimed that the animals would perform at 80% and not significantly less. We checked this throughout the testing protocols as well, where we also presented the old pair as quality control, and their performance was never significantly lower than our 80% performance threshold on this pair, suggesting that they mastered the task in which they were trained. To avoid discussion on semantics, we also rephrased “master the task” into “learn the task”.

• What are the criteria for the claim that the "animal model of choice for vision studies has become the rodent model"? It is likely that researchers in primate vision may hold a different viewpoint, and data such as yearly total publication counts might not align with this claim.

Primate vision is important for investigating complex visual aspects. With the advancements in experimental techniques for rodent vision, e.g. genetics and imaging techniques as well as behavioural tasks, the rodent model has become an important model as well. It is not necessarily an “either” or “or” question (primates or rodents), but more a complementary issue: using both primates and rodents to unravel the full picture of vision.

We have changed this part in the introduction to “Lately, the rodent model has become an important model in vision studies, motivated by the applicability of molecular and genetic tools rather than by the visual capabilities of rodents”.

• The correspondence between the list of layers in Supplementary Tables 8 and 9 and the layers shown in Figures 4 and 6 could be clarified.

We have clarified this in the caption of Figure 7

• The titles in Figures 4 and 6 could be updated from "DNN" to "cDNN" to ensure consistency with the rest of the manuscript.

Thank you for your feedback. We have changed the titles in Figures 4 and 6 such that they are consistent with the rest of the manuscript.